# REASONGEN-R1: COT FOR AUTOREGRESSIVE IMAGE GENERATION MODEL THROUGH SFT AND RL

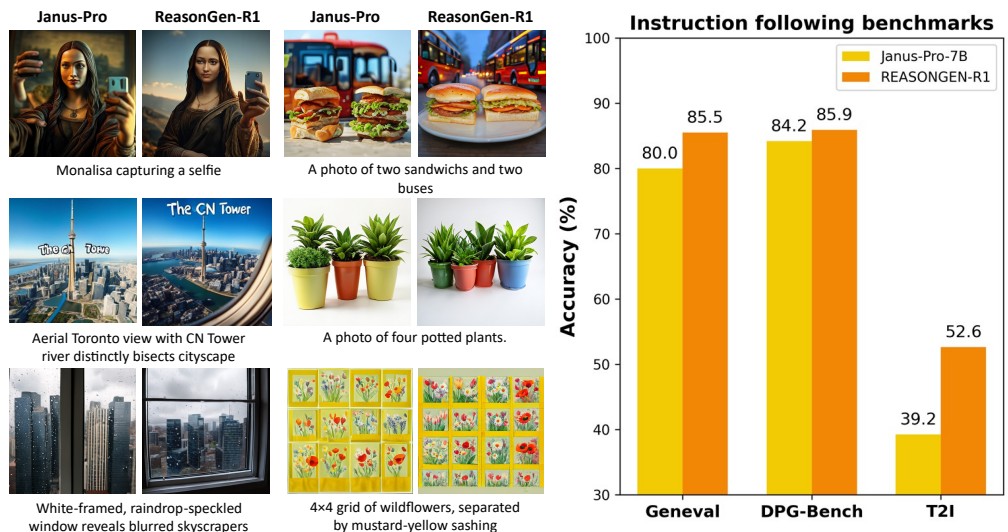

Figure 1: Left: We show side-by-side visualizations of images generated by Janus-Pro-7B and REASONGEN-R1 using identical prompts (prompts are summarized; see the raw prompts in Table 1, 2). Right: we present a performance comparison across three instruction-following benchmarks. In every benchmark, REASONGEN-R1 outperforms the base Janus-Pro-7B model, demonstrating a substantial improvement in its ability to follow instructions.

## ABSTRACT

Although chain-of-thought (CoT) reasoning and reinforcement learning (RL) have driven breakthroughs in large language models(LLMs), their integration into generative vision models remains underexplored. We introduce ReasonGen-R1, a two-stage framework that first imbues an autoregressive image generator with explicit text-based "thinking" skills via supervised fine-tuning (SFT) on a newly generated reasoning dataset of written rationales, and then refines its outputs using Group Relative Policy Optimization (GRPO). To enable the model to reason through text before generating images, We automatically generate and release a corpus of model-crafted rationales paired with input prompts, enabling controlled planning of object layouts, styles, and scene compositions. Our GRPO algorithm uses reward signals from a pretrained vision–language model to assess overall visual quality, optimizing the policy in each update. We further design an adaptive entropy loss to prevent model collapse in this relatively complex task. Evaluations on GenEval, DPG, and the T2I benchmark demonstrate that ReasonGen-R1 consistently outperforms strong baselines and prior state-of-the-art models.

## 1 INTRODUCTION

Recent breakthroughs in models such as OpenAI's o1 (OpenAI, 2024) and Deepseek's R1 (DeepSeek-AI et al., 2025) have demonstrated the significant advantages of reinforcement learn-

ing (RL) methods for enhancing the thinking and reasoning capabilities of large language models (LLMs). These advancements confirm that step-by-step reasoning substantially improves answer accuracy and robustness. Naturally, transferring the powerful reasoning capabilities of LLMs from text-based tasks to image generation tasks becomes critically important. Models such as Janus-Pro (Chen et al., 2025b) and X-Omni (Geng et al., 2025) have introduced unified image-generation paradigms, highlighting that multimodal LLM-based autoregressive content generation exhibits superior instruction-following abilities and image quality. Consequently, a crucial next step is exploring how to effectively incorporate thinking and reasoning via RL into autoregressive generation models.

Motivated by human creative processes, where artists typically contemplate structural and sequential considerations before image creation, we aim to enable autoregressive generation models to autonomously produce textual reasoning sequences based on user prompts. This approach harnesses the inherent instruction-following strengths of autoregressive image generation models, encouraging them to make creative associations and structural decisions akin to human thought processes. DALL·E 3 (OpenAI, 2023) enhances image quality by having ChatGPT automatically generate tailored, detailed prompts when a user provides an input query. For recent unified generation–understanding model like Janus-Pro, we aim to realize this coarse-to-fine think-and-generate process in an end-to-end manner by first producing a textual chain of thought (CoT) followed by image generation. Furthermore, we employ RL to jointly optimize both the quality of the textual CoT and the resulting images.

However, achieving this goal introduces several challenges. First, current autoregressive image-generation models, such as Janus-Pro and X-Omni, typically generate images directly from text prompts without the ability to concurrently produce textual reasoning. Consequently, straightforward RL supervision might be ineffective. Second, designing an efficient RL-based post-training pipeline that facilitates "thinking-based" generation within autoregressive image-generation frameworks remains unexplored and unvalidated.

To address these challenges, we introduce REASONGEN-R1, a novel two-stage training paradigm combining supervised fine-tuning (SFT) with chain-of-thought (CoT) and RL using Group Relative Policy Optimization (GRPO) (Li et al., 2025a), tailored explicitly for pretrained autoregressive image-generation models. Our training objective is first to adapt the model to a reasonable text reasoning distribution for images, and then to use RL to guide the model toward recognizing which reasoning paths can lead to better generation quality. To overcome the initial challenge of familiarizing the model with reasonable reasoning trajectories, we employ an "instruction → CoT → image" pipeline to jointly supervise textual reasoning sequences and image outputs during SFT training. We constructed a comprehensive dataset comprising 200k samples from the LAION aesthetics subset (Schuhmann et al., 2022), meticulously annotated using GPT-4.1 (OpenAI, 2025) to include rich CoT reasoning trajectories for each `<instruction, CoT, image>` triplet, covering diverse reasoning scenarios. This dataset effectively balances the dual objectives of CoT text generation and target image generation without compromising image quality. For the second challenge—teaching the model which CoT trajectories effectively promote generation quality—REASONGEN-R1 adopts an an efficient GRPO framework utilizing the powerful image-understanding model Qwen-2.5-VL (Bai et al., 2025) as the reward model. For each training rollout, we assess prompt–image alignment by querying Qwen-2.5-VL for a binary consistency score. Additionally, we found reinforcement lwearning with interleaved modality output extremely sensitive to entropy explosion or entropy vanishing. To tackle this, we introduce a unique adaptive entropy loss design to ensure stable and effective training.

Empirical results in Figure 1 demonstrate that REASONGEN-R1 can significantly improve the quality of generated images, and achieves superior performance on benchmark datasets such as GenEval (Ghosh et al., 2023) (+6%), DPG-Bench (Hu et al., 2024) (+1.69%), and T2I-Benchmark (Huang et al., 2023) (+13.38%), thereby substantially enhancing reasoning-based generation capabilities. Our contributions are threefold:

1. We are the first to integrate the reasoning process into autoregressive image generation via a two-stage SFT + GRPO training framework, establishing a strong baseline for future "think-and-generate" content creation.

2. Technically, we build a large-scale CoT image-generation dataset to spark the model's exploration of reasoning, then leverage multimodal LLMs as reward models within GRPO to guide the model toward more suitable reasoning for I2T tasks; an adaptive entropy loss further mitigates entropy explosion or collapse.

3. Extensive experiments confirm the efficiency and effectiveness of the proposed REASONGEN-R1 framework across multiple benchmarks.

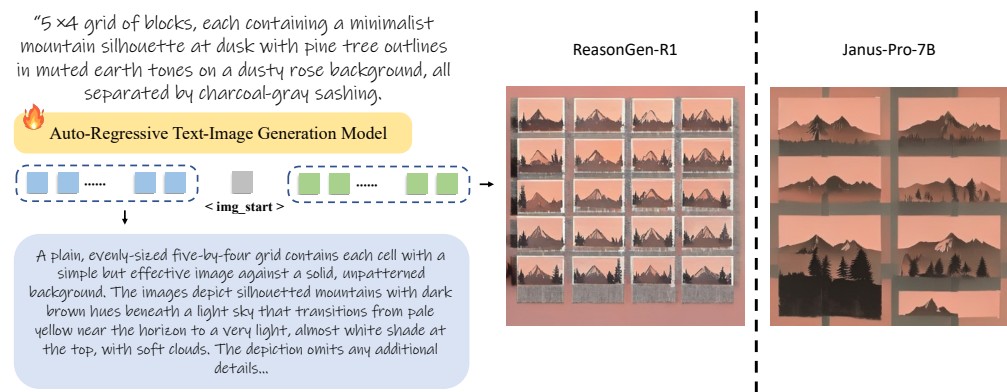

Figure 2: Overall framework of *ReasonGen-R1*. We propose the first reinforcement learning post-training framework that enables autoregressive image generation models to output both a chain-of-thought reasoning process and the final image.

## 2 RELATED WORK

### 2.1 UNIFIED AUTO-REGRESSIVE GENERATION MODELS

Recent work on unified auto-regressive generation models (Xie et al., 2024b; Wang et al., 2024b; Dong et al., 2023; Li et al., 2024a; Tong et al., 2024; Fang et al., 2024) has demonstrated that a single model can both generate language and images. By mapping images into the same embedding space as text and feeding both into an LLM backbone, these systems attain a richer understanding of prompts and can produce multimodal outputs. However, most existing designs—such as Janus-Pro (Chen et al., 2025b)—still generate text and images in distinct phases by default. This modality split prevents true interleaving of words and pixels in one continuous sequence, limiting their effectiveness on tasks that demand integrated, cross-modal reasoning.

### 2.2 CHAIN-OF-THOUGHT IN LLMS

Chain-of-thought (CoT) reasoning has emerged as an effective strategy in large language models (LLMs), allowing models to decompose complex tasks into intermediate logical steps. This technique has led to state-of-the-art results in mathematical problem-solving, commonsense reasoning, and compositional tasks in models such as PaLM (Chowdhery et al., 2022), GPT-4 (OpenAI et al., 2023), and LLaMA 2 (Touvron et al., 2023). Wei et al. (Wei et al., 2023) and subsequent works have shown that reasoning traces not only improve model performance but also enable interpretability and controllability.

### 2.3 REINFORCEMENT LEARNING IN LLMS AND LVLMS REASONING

More recently, reinforcement learning (RL) has been employed to improve the reasoning capabilities of LLMs and vision-language models (VLMs). For instance, leading by DeepSeek-R1 (DeepSeek-AI et al., 2025), a number of works (Meng et al., 2025; Yang et al., 2025; Zhang et al., 2025; Yu et al., 2025; Deng et al., 2025; Li et al., 2025a;b) utilizes Group Relative Policy Optimization (GRPO) and its variants, to enhance its reasoning ability without the need for a separate critic model.

This approach normalizes rewards within a group of generated outputs, reducing computational cost and improving performance.

Despite reinforcement learning (RL) has been increasingly applied to refine image generation models, particularly in text-to-image synthesis (Wallace et al., 2023; Lee et al., 2024; Wei et al., 2024; Oertell et al., 2024; Fan et al., 2023), the integration of RL specifically for reasoning within image generation remains largely unexplored. In this work, we explore the possibility of the magical reasoning ability for enhanced image generation.

## 3 METHOD

REASONGEN-R1 consists of two main parts: (1) supervised finetuning on a base autoregressive image generation model, equipping the model with textual reasoning ability. (2) reinforcement learning on the finetuned model, further enhancing the model's capability to analyze the prompt and output the final image.

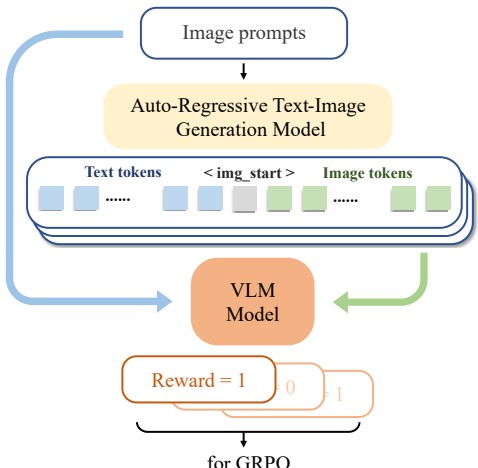

Figure 3: The pipeline for reinforcement learning in *ReasonGen-R1*.

### 3.1 SUPERVISED FINETUNING

#### 3.1.1 DATASET CONSTRUCTION

To equip the base model with the ability to generate interleaved text-image outputs, we begin by constructing a diverse dataset consisting of short prompts, dense prompts, and corresponding images. Inspired by DALL·E 3 (OpenAI, 2023), we adopt a coarse-to-fine think-and-generate pipeline to enhance image generation quality, thereby enabling more effective performance on text-to-image tasks. Specifically, we first select 200,000 images from the LAION-Aesthetic V1 dataset, which is a subset of the LAION-5B collection (Schuhmann et al., 2022). Since the base model, Janus-Pro, is restricted to generating square images, we crop the long side of each image to match the shorter side, resulting in a square output.

Next, we query GPT-4.1 (OpenAI, 2025) to generate a concise caption for each image, focusing on key details such as object color, counts, spatial relationships, and other contextual elements. Then, we use GPT-4.1 to augment the concise caption, generating additional prompts to increase diversity. These augmented prompts include a set of image tags, object-centric phrases, three paraphrased versions of the concise caption, and one varied caption written in a different style. At the same time, we also generate a detailed caption, providing a longer, more comprehensive description of the image. Notably, the concise caption is generated directly from the image input, while the augmented prompts and detailed caption are generated solely from the concise caption. This approach ensures that GPT-4.1 does not introduce additional information from the image, preventing potential information discrepancies during the SFT training as the generation model being trained only has access to the concise caption. For details on the data construction process, the system prompt used to call GPT, and the strategies for augmenting the concise prompt, please refer to Appendix A.1.

### 3.1.2 SFT TRAINING

Prior works such as Hu et al. (2025) and Chen et al. (2025a) have clearly demonstrated the critical role of SFT cold-start in RL. It is essential to first familiarize the model with the distribution of suitable textual CoT for image generation tasks; only then can subsequent RL effectively learn which reasoning trajectories truly enhance performance. In fact, unified models such as Janus-Pro and X-Omni cannot even output text alongside images during generation, since they have not been exposed to this type of interleaved training—let alone acquire the ability of reasoning for generation

tasks. Therefore, we require SFT cold-start training to enable this capability before proceeding with the subsequent RL process.

In our supervised fine-tuning (SFT) stage, we address this limitation by training the model to first generate a coherent reasoning rationale and then seamlessly transition to image synthesis within a single sequence. We build on Janus-Pro-7B as the base model, which uses a special image-start token to trigger visual output. As illustrated in Figure 2, each training sequence begins with a concise image prompt followed by the detailed reasoning caption. We then insert the image-start token and the corresponding image tokens. Through this formatting, the model learns to produce a detailed rationale before autonomously emitting the image-start token and generating the final image.

## 3.2 REINFORCEMENT LEARNING

Group Relative Policy Optimization (GRPO) (Shao et al., 2024) has shown a strong capability to explore the reasoning potential of LLMs. To further align generated images with the text-based rationale and input prompt, we adapt Group Relative Policy Optimization (GRPO) for image generation as in Figure 3.

### 3.2.1 REINFORCEMENT LEARNING ALGORITHM

**GRPO** computes advantages relative to a group of responses. For each question–answer pair $(q, a)$, let the old policy $\pi_{\theta_{\text{old}}}$ sample $G$ responses $\{o_i\}_{i=1}^G$, each yielding reward $R_i$. Then, normalize it to obtain

$$\hat{A}_{i,t} = \frac{r_i - \text{mean}\big(\{R_i\}_{i=1}^G\big)}{\text{std}\big(\{R_i\}_{i=1}^G\big)}. \tag{1}$$

The GRPO policy objective then mirrors PPO's clipped surrogate, plus a KL penalty:

$$\mathcal{J}_{\text{GRPO}}(\theta) = \mathbb{E}_{(q,a)\sim\mathcal{D},\,\{o_i\}_{i=1}^G \sim \pi_{\theta_{\text{old}}}(\cdot|q)}$$

$$\left[ \frac{1}{G} \sum_{i=1}^G \frac{1}{|o_i|} \sum_{t=1}^{|o_i|} \min\big(r_{i,t}(\theta)\,\hat{A}_{i,t},\, \text{clip}\big(r_{i,t}(\theta),\, 1-\varepsilon,\, 1+\varepsilon\big)\,\hat{A}_{i,t}\big) - \beta\,D_{\text{KL}}\big(\pi_\theta \| \pi_{\text{ref}}\big) \right]. \tag{2}$$

Finally, each per-token importance weight is

$$r_{i,t}(\theta) = \frac{\pi_\theta\big(o_{i,t} \mid q,\, o_{i,<t}\big)}{\pi_{\theta_{\text{old}}}\big(o_{i,t} \mid q,\, o_{i,<t}\big)}. \tag{3}$$

Beyond GRPO, our RL algorithm adds an adaptive entropy loss to further enhance the stability of the training.

**Adaptive entropy loss** is inspired by SAC (Haarnoja et al., 2018; 2019), a RL algorithm that is commonly used in Robotics RL. More specifically, adaptive entropy loss sets a target entropy. And during training, the entropy loss coefficient would automatically update through gradient descent. The adaptive entropy loss function in Haarnoja et al. (2019) is:

$$\mathcal{L}_\alpha = \mathbb{E}_{a_t \sim \pi}\left[ \alpha \cdot \big(\log \pi(a_t|s_t) + \mathcal{H}_{\text{target}}\big) \right], \quad \alpha = \log(\phi) \tag{4}$$

where the $\alpha$ is the learnable entropy loss term, $\mathcal{H}_{\text{target}}$ stands for the target entropy and $\log \pi(a_t|s_t)$ stands for the entropy of the current action output conditioned on the current state in robot learning. The learnable entropy loss term $\alpha$, often parameterized as $\log(\phi)$, is always positive, encouraging the model to explore more.

In our setting, we found that the model is very prone to both entropy vanishing and entropy explosion, often leading to mode collapse in image generation. To tackle this, we modify the parameterization method of $\alpha$ to $\arcsin(\phi)$, so that we can both learn positive and negative $\alpha$. The complete loss term for updating $\alpha$ is:

$$L_\alpha = \mathbb{E}_{a \sim \pi}\left[ \alpha \cdot \big(\log \pi(o_{i,t}|q, o_{i<t}) + \mathcal{H}_{\text{target}}\big) \right], \quad \alpha = \arcsin(\phi) \tag{5}$$

In addition, our RL algorithm uses batch sub-sampling to remove groups that score all 0 or all 1 and remove the KL loss. Our final objective function is therefore written as:

$$
\mathcal{J}(\theta) = \mathbb{E}_{(q,a)\sim\mathcal{D},\, \{o_i\}_{i=1}^G \sim \pi_{\theta_{\text{old}}}(\cdot|q)}
$$

$$
\left[ \frac{1}{G} \sum_{i=1}^G \frac{1}{|o_i|} \sum_{t=1}^{|o_i|} \min\left( r_{i,t}(\theta)\hat{A}_{i,t}, \text{clip}\left(r_{i,t}(\theta), 1-\varepsilon, 1+\varepsilon\right)\hat{A}_{i,t} \right) + \alpha \log \pi(o_{i,t}|q, o_{i<t}) \right]
\tag{6}
$$

### 3.2.2 REWARD DESIGN

As opposed to rule-based reward in original GRPO in LLM training, it's difficult to have golden pre-defined rules to assess the consistency between prompt and generated image. To tackle this, we use a strong VLM, Qwen-2.5-VL 7B (Bai et al., 2025), as our reward model. The specific system prompt used for the reward model is provided in Appendix A.3. In each rollout, the generation model produces a single sequence: `prompt` → `reasoning` → `image`. We compute binary rewards solely on the generated image by querying a pretrained vision–language model (VLM) to evaluate consistency between the input text and output image. To credit the preceding reasoning steps, we propagate the image-level reward back through the entire sequence, reinforcing textual rationales that yield higher-quality visual outputs.

## 4 EXPERIMENT

In our experiment section, we aim to answer the following questions:(1) To what extent does incorporating textual reasoning improve instruction adherence in image generation? (2) How much does the RL benefit from the SFT training warmup? (2) How much does the RL training benefit from the size of reward model? (4) How does the adaptive entropy loss benefit the training?

### 4.1 EXPERIMENT SETUP

#### 4.1.1 EVALUATION BENCHMARKS

**T2I Benchmark (Huang et al., 2023):** 6,000 compositional prompts spanning attribute binding, object relations, and complex scene layouts (color, shape, texture bindings; spatial and non-spatial relations).

**GenEval (Ghosh et al., 2023):** Object-focused alignment tasks designed to assess fine-grained consistency between text and image outputs.

**DPG-Bench (Hu et al., 2024):** Dense-prompt generation emphasizing detailed instructions.

These benchmarks collectively cover a wide spectrum of compositional and alignment challenges, providing a thorough evaluation of the model's ability to follow complex textual directives.

#### 4.1.2 DATASET SETTINGS

Our training dataset comprises prompts synthesized from three benchmarks: GenEval, DPG-Bench, and T2I-CompBench++ (Huang et al., 2025). For GenEval, we enlarge its object vocabulary from 80 to 308 and extend its original generator with a new variant that specifies two distinct objects along with their respective counts. Using this augmented generator, we synthesized 12,552 unique prompts and filtered out any that overlap with the GenEval test set, resulting in 12367 prompts. For the DPG-Bench, we leveraged GPT-4.1 to produce 5,000 fresh prompts: for each draft, we sampled five existing DPG prompts at random and instructed GPT-4.1 to craft a final prompt matching their length and style. Lastly, we incorporated all prompts from the official T2I-CompBench++ training split without modification, resulting in 11,003 prompts. Notice that we ensure there's no overlap between synthesized training set and the original test sets.

Table 1: Quantitative comparison results on the GenEval (Ghosh et al., 2023) benchmark. Ours shows significant improvement over previous methods.

| Method | Single Obj. | Two Obj. | Counting | Colors | Position | Color Attri. | Overall ↑ |
|---|---|---|---|---|---|---|---|
| *Diffusion Models* | | | | | | | |
| SD-1.5 Rombach et al. (2022) | 0.97 | 0.38 | 0.35 | 0.76 | 0.04 | 0.06 | 0.43 |
| PixArt-$\alpha$ Chen et al. (2023) | 0.98 | 0.50 | 0.44 | 0.80 | 0.08 | 0.07 | 0.48 |
| SDXL-base-1.0 Podell et al. (2023) | 0.98 | 0.74 | 0.39 | 0.85 | 0.15 | 0.23 | 0.55 |
| DALL·E-3 OpenAI (2023) | 0.96 | 0.87 | 0.47 | 0.83 | 0.43 | 0.45 | 0.67 |
| SD3-Medium Esser et al. (2024) | 0.99 | 0.94 | 0.72 | 0.89 | 0.33 | 0.60 | 0.74 |
| *Autoregressive Models* | | | | | | | |
| Show-o Xie et al. (2024a) | 0.95 | 0.52 | 0.49 | 0.82 | 0.11 | 0.28 | 0.53 |
| Emu3 Wang et al. (2024b) | 0.98 | 0.71 | 0.34 | 0.81 | 0.17 | 0.21 | 0.54 |
| D-DiT Li et al. (2024b) | 0.97 | 0.80 | 0.54 | 0.76 | 0.32 | 0.50 | 0.65 |
| ILLUME Wang et al. (2024a) | 0.99 | 0.86 | 0.45 | 0.71 | 0.39 | 0.28 | 0.61 |
| Janus-Pro-7B (Baseline) Chen et al. (2025b) | 0.99 | 0.89 | 0.59 | 0.90 | 0.79 | 0.66 | 0.80 |
| **REASONGEN-R1 (Ours)** | **0.99** | **0.94** | **0.62** | **0.90** | **0.84** | **0.84** | **0.86** |

Table 2: Quantitative comparison results on the DPG-Bench (Hu et al., 2024).

| Model | Global | Entity | Attribute | Relation | Other | Overall ↑ |
|---|---|---|---|---|---|---|
| SD-1.5 Rombach et al. (2022) | 74.63 | 74.23 | 75.39 | 73.49 | 67.81 | 63.18 |
| PixArt-$\alpha$ Chen et al. (2023) | 74.97 | 79.32 | 78.60 | 82.57 | 76.96 | 71.11 |
| SDXL Podell et al. (2023) | 83.27 | 82.43 | 80.91 | 86.76 | 80.41 | 74.65 |
| DALL·E-3 OpenAI (2023) | 90.97 | 89.61 | 88.39 | 90.58 | 89.83 | 83.50 |
| SD3-Medium Esser et al. (2024) | 87.90 | 91.01 | 88.83 | 80.70 | 88.68 | 84.08 |
| Janus-Pro-7B (Baseline) Chen et al. (2025b) | 86.90 | 88.90 | 89.40 | 89.32 | **89.48** | 84.19 |
| **REASONGEN-R1 (Ours)** | **91.66** | **90.92** | **90.72** | **90.62** | 87.49 | **85.88** |

### 4.1.3 TRAINING SETTINGS

In all supervised fine-tuning (SFT) experiments, we trained the model for 1 epoch and selected the final checkpoint. For reinforcement learning (RL) experiments, we trained the model for 300 steps and chose the checkpoint with the highest validation reward.

### 4.2 MAIN RESULT

To answer the first question: To what extent does incorporating textual reasoning improve instruction adherence in image generation? We compare REASONGEN-R1 against the Janus-Pro-7B baseline and leading diffusion and auto-regressive text-to-image systems across three diverse benchmark suites. As shown in Table 1, 2, 3 REASONGEN-R1 outperforms the base model in all three benchmarks. REASONGEN-R1 also surpasses many leading image-generation models. These results indicate that the textual-reasoning model framework and SFT-RL pipeline greatly boost the performance of the auto-regressive image generation model.

We analyze numerous CoT outputs produced by our model in Appendix B.1. Overall, the CoT generated by REASONGEN-R1 effectively steers the model to produce more reasonable outputs, as anticipated. In Appendix B.3, we provide additional analyses of image generation quality, showing that REASONGEN-R1 significantly improves the visual quality of the generated images.

### 4.3 ABLATION STUDY

### 4.3.1 EFFECTIVENESS OF SFT TRAINING

To answer the second research question: How much does the RL benefit from the SFT training warmup? We compared REASONGEN-R1 with pure RL. Since the original model doesn't have the capability to control its output modality during RL rollout, we first instruct the model to output reasoning text. Once it finishes text generation, we swap its last end of sentence token with an image start token to start image generation.

Table 3: Quantitative comparison results on the T2I-Benchmark (Huang et al., 2023).

| Model | Attribute Binding ↑ | | | Object Relationship ↑ | | Complex ↑ |
|---|---|---|---|---|---|---|
| | Color | Shape | Texture | Spatial | Non-Spatial | |
| *Diffusion Models* | | | | | | |
| PixArt-$\alpha$ Chen et al. (2023) | 0.6690 | 0.4927 | 0.6477 | 0.2064 | 0.3197 | 0.3433 |
| CoMat Jiang et al. (2024) | 0.7827 | 0.5329 | 0.6468 | 0.2428 | 0.3187 | 0.3680 |
| SD-1.5 Rombach et al. (2022) | 0.3758 | 0.3713 | 0.4186 | 0.1165 | 0.3112 | 0.3047 |
| SDXL-base-1.0 Podell et al. (2023) | 0.5879 | 0.4687 | 0.5299 | 0.2131 | 0.3119 | 0.3237 |
| SD3 Esser et al. (2024) | 0.8132 | 0.5885 | 0.7334 | 0.3200 | 0.4084 | 0.3771 |
| DALL·E-3 OpenAI (2023) | 0.7785 | 0.6205 | 0.7036 | 0.2865 | 0.3744 | 0.3773 |
| FLUX.1 Labs (2024) | 0.7407 | 0.5718 | 0.6922 | 0.2863 | 0.3127 | 0.3703 |
| *AutoRegressive Models* | | | | | | |
| Show-o Xie et al. (2024a) | 0.56 | 0.41 | 0.46 | 0.20 | 0.30 | 0.29 |
| Show-o Xie et al. (2024a) + PARM Guo et al. (2025) | 0.75 | 0.56 | 0.66 | 0.29 | 0.31 | 0.37 |
| Emu3 Wang et al. (2024b) | 0.7544 | 0.5706 | 0.7164 | – | – | – |
| Janus-Pro-7B (Baseline) Chen et al. (2025b) | 0.6359 | 0.3528 | 0.4936 | 0.2061 | 0.3085 | 0.3559 |
| **REASONGEN-R1 (Ours)** | **0.8321** | **0.565** | **0.7295** | **0.3007** | **0.3375** | **0.3909** |

Table 4: Ablation Study on SFT stage, RL stage and Reward model size. *w/o SFT* and *w/o RL* means only single stage is applied. *w/ Small Rewarder* means changing 7B reward model into 3B.

| Method | Rewarder Size | Single Obj. | Two Obj. | Counting | Colors | Position | Color Attri. | Overall ↑ |
|---|---|---|---|---|---|---|---|---|
| **REASONGEN-R1 (Ours)** | 7B | **0.99** | **0.94** | **0.62** | **0.90** | **0.84** | **0.84** | **0.86** |
| w/o SFT | 7B | 0.99 | 0.86 | 0.29 | 0.84 | 0.45 | 0.65 | 0.68 |
| w/o RL | - | 0.86 | 0.64 | 0.31 | 0.75 | 0.40 | 0.46 | 0.57 |
| w/o Adaptive Entropy Loss | 7B | 0.99 | 0.89 | 0.47 | 0.87 | 0.45 | 0.66 | 0.72 |
| w/ Small Rewarder | 3B | 0.50 | 0.48 | 0.39 | 0.61 | 0.41 | 0.32 | 0.45 |

As shown in Table 4, REASONGEN-R1 significantly outperforms the *w/o SFT* baseline by 18%, indicating that SFT primes the base model to carry out proper interleaved reasoning and generation. The *w/o RL* variant, however, reveals that SFT alone is not sufficient, because the GPT-annotated CoT traces do not always represent the reasoning trajectories most conducive to high-quality image synthesis. Nevertheless, the gap between *w/o SFT* and *w/o RL* shows that SFT equips the model to explore diverse thinking paths; the subsequent RL stage is therefore essential to unlock this potential, fully realize the "think-and-generate" motivation, and achieve the final performance gains.

### 4.3.2 REWARD MODEL SIZE MATTERS

For our third research question, we use Qwen-2.5-VL-3B as the rewarder VLM for comparison. In Table 4, a smaller VLM failed to provide good reward signals, leading to poorer performance after RL. This indicates that using a large and accurate rewarder model is crucial to our RL algorithm.

### 4.3.3 STABLE TRAINING WITH ADAPTIVE ENTROPY LOSS

To address our final research question, we perform a comparative analysis of our model under two configurations. The first configuration involves RL training the model without any entropy loss, allowing the model to learn without any explicit regularization on entropy. The second configuration incorporates a fixed entropy loss term, where the entropy of the model's predictions is penalized by a constant value during training. This setup allows us to evaluate the impact of adaptive entropy regularization on the model's performance and the stability of RL training.

### 4.3.4 ABLATION ON REASONING STYLE

In this work, we adopt by default a DALL·E 3-style coarse-to-fine reasoning approach. Here, we additionally compare it with a step-by-step reasoning style that is closer to OpenAI's O1 paradigm. This setup is similar to the coarse-to-fine scheme, except that the reasoning traces in the SFT dataset are replaced with step-by-step reasoning. As shown in Table 5, the experimental results indicate that the overall performance difference between the two styles is small. However, there are task-specific

variations: step-by-step reasoning performs slightly better on tasks such as Colors and Position, while the coarse-to-fine approach clearly outperforms on Counting and Color Attri..

Table 5: Comparison of different reasoning style on GenEval benchmark.

| Reasoning Style | Single Obj. | Two Obj. | Counting | Colors | Position | Color Attri. | Overall |
|---|---|---|---|---|---|---|---|
| Coarse-to-Fine | **0.99** | **0.94** | **0.62** | 0.90 | 0.84 | **0.84** | **0.86** |
| Step-by-Step REASONGEN-R1 | **0.99** | **0.94** | 0.60 | **0.93** | **0.87** | 0.80 | **0.86** |

### 4.3.5 BINARIZED REWARDS IMPROVE LEARNING

Table 6: Comparison of binary and non-binary reward designs on GenEval Benchmark.

| Reward Design | Single Obj. | Two Obj. | Counting | Colors | Position | Color Attri. | Overall |
|---|---|---|---|---|---|---|---|
| Binary | **0.99** | **0.94** | **0.62** | **0.90** | **0.84** | **0.84** | **0.86** |
| Non-Binary Score | **0.99** | 0.87 | 0.51 | 0.86 | 0.66 | 0.70 | 0.77 |

Our reward design follows the GRPO algorithm, which employs a binary reward. For comparison, we also evaluate a non-binary variant, where the reward model (Qwen-7B) outputs an integer score from 1 to 5, which is then normalized to 0 to 1. As shown in Table 6, the non-binary reward variant performs worse than the binary version. We attribute this to the increased noise and difficulty of reliably generating fine-grained scores in a zero-shot VLM setting.

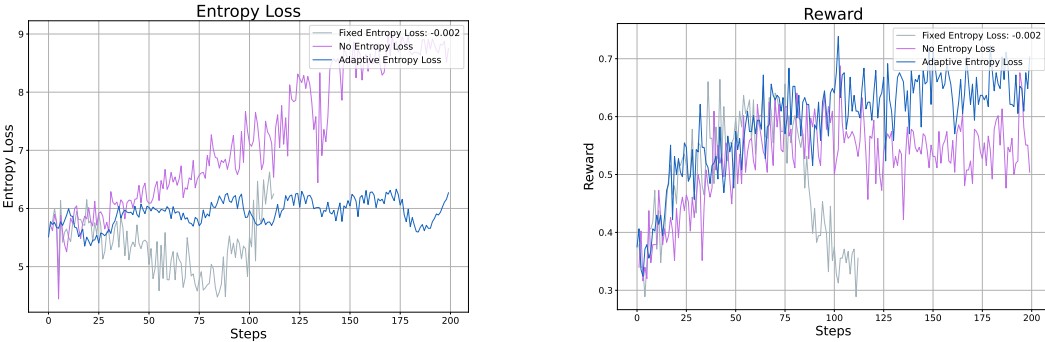

Figure 4: Comparison on Entropy Loss regularization.

As illustrated in Figure 4, RL without entropy loss experiences entropy explosion after 100 training steps, resulting in degraded performance. On the other hand, applying a fixed entropy penalty of -0.002 causes a continuous decline in entropy, reaching dangerously low levels at step 80, which leads to mode collapse and a sharp decrease in reward. These observations highlight the challenges of RL training with interleaved text and images, particularly its sensitivity to entropy loss regularization. In contrast, our adaptive entropy loss effectively maintains entropy within an optimal range, ensuring stable training. As shown in Table 4, RL without entropy loss got a significant performance drop.

## 5 CONCLUSION

In this paper, we propose REASONGEN-R1, a two-stage framework that integrates Chain-of-Thought (CoT) reasoning with reinforcement learning (RL) to enhance autoregressive image generation. By combining supervised fine-tuning with GRPO-based RL, our method enables interleaved reasoning and image synthesis, yielding better instruction adherence and higher visual quality.

Experiments on GenEval, DPG-Bench, and T2I-Benchmark show that REASONGEN-R1 surpasses Janus-Pro and other state-of-the-art models. Our study also highlights the necessity of a strong reward model and adaptive entropy loss for stable training. We will release our dataset, models, and code to support future research.

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

# A METHOD DETAILS

## A.1 DATASET CONSTRUCTION DETAILS

Our dataset construction involves three distinct calls to the OpenAI API, each serving a different role:

**1. Concise Image Captioning** We use GPT-4.1 mini to generate a short, accurate, and informative caption that highlights object counts, colors, positions, and other details. During this stage, we feed the GPT with image.

```
You are a data annotation expert.  Generate a concise
image caption for the image, focusing specifically
on the color, number, position, and other details
```

```
of objects, background, and humans present.  Analyze
carefully and ensure accuracy in your description.
The caption should be a single short sentence that
faithfully includes most of the important information
in the original caption.
```

**2. Concise Image Caption Augmentation** We use GPT-4.1 Nano to augment the concise caption obtained from the previous API call. We augment the caption into several categories. The purpose for the augmentation is to prevent the model from overfitting to one specific prompt pattern during the SFT training. During this stage, we don't give the image to the GPT, concise caption from the previous call would be the only input.

```
You are an image-annotation augmentation expert.
Given the original detailed caption below, analyze
it and produce a single JSON object (and only that
JSON) with the following top-level keys|no nested
structures:
• \concise_caption":  A one-sentence compressed
caption capturing the main objects, their colors and
positions.
• \paraphrases":  An array of 3 alternative
one-sentence phrasings that preserve every key
details but vary word order and synonyms.
• \tags":  An array of 5{8 keywords describing
objects, colors, positions, and scene.
• \varied_captions":  An array of 3 one-sentence
captions, each in a different style of your choice
(the model should decide the styles randomly).
• \object_prompts":  An array of several very short
prompts in the form \a/an/number optional adj.  n.",
listing only the main object noun (no more than 3)
in descending order of their significance.(e.g.  \a
clock", \two wooden chairs").
**Input (detailed caption):**
\{CONCISE_CAPTION}"
**Expected Output (json):**
{
"concise_caption":  "...",
"paraphrases":  ["...", "...", "..."],
"tags":  ["...", "...", "...", "...", "..."],
"varied_captions":  ["...", "...", "..."],
"object_prompts":  ["...", "..."]
}
```

**3. Detailed Caption Generation** We use GPT-4.1 Nano to generate a detailed caption from each concise caption. This detailed caption serves as the ground truth chain-of-thought (CoT) supervision during the supervised fine-tuning (SFT) stage, guiding the model to learn reasoning based solely on the concise prompt. Importantly, GPT is provided only with the concise caption and not the corresponding image when generating the detailed caption. This design choice ensures that no additional visual information leaks into the supervision, preventing an information gap during SFT—where the model only has access to the concise caption. Without this precaution, the model may learn to generate overly imaginative or irrelevant reasoning that is not grounded in the available input.

```
You are an image-annotation augmentation expert.
Given the following inputs:
Input \concise_caption":  A concise description of
the image (e.g.  \A red clock on a wooden table")
• \expanded_prompt":  A richly detailed prompt that
(1) restates the concise_caption with full color,
count, position, background, and mood.
**Inputs**:
```

```
        concise_caption = "{concise_caption}"
        **Output** (Please directly output the
        expanded_prompt):
```

## A.2 SFT Training Details

**Prompt Augmentation** To prevent overfitting to a single prompt format, we apply prompt augmentation strategies described in Appendix A.1 to enhance prompt diversity. Specifically, during training, we uniformly sample one augmentation type (treating the original concise caption as one of the types) to replace the original prompt. If the selected type is *tags* or *object_prompts*, we concatenate all items using commas. If the type is *paraphrases* or *varied_captions*, we randomly select one candidate from the list. If *concise_caption* is selected, we use it directly without modification.

**Prompt Formating** To reduce the distribution gap between the base model and our target model, we add a bridging prompt between the concise prompt and the ground truth CoT. More specifically, we add the following:

```
        Output a richly detailed prompt:
```

## A.3 RL Algorithm Details

**Adaptive Entropy Loss** To stabilize reinforcement learning with interleaved text and image outputs, we employ an Adaptive Entropy Loss Haarnoja et al. (2019). We have two independent Adaptive Entropy Loss regularizers for each modality. Because image and text tokens have vastly different vocabulary sizes—and consequently different natural entropy scales—we maintain separate entropy targets for each. Specifically, we use a target entropy of 7.0 for image tokens and 2.0 for text tokens. These target entropy values come from the average rollout entropy observed after supervised fine-tuning.

**Reward Model** We use Qwen-2.5-VL-7B Bai et al. (2025) as our reward VLM model for reinforcement learning. During training, we'll prompt it with the following template for each rollout image:

```
        You are given a text prompt:  "{prompt}"
        Below is one generated image:  <image>
        1.  Describe the image thoroughly (objects, colors,
        layout, etc.), do not be affected by the prompt.
        2.  Identify key visual elements and instructions from
        the prompt.
        3.  Evaluate how well the image follows the prompt:
        - Are all required elements present?
        - Are object counts, colors, and positions accurate?
        Be extremly strict and precise:
        Only if the image matches the prompt perfectly,
        respond with:  \boxed{1}.
        Otherwise, respond with:  \boxed{0}
        Reason before your final boxed answer.  Only one
        number should appear inside the box.
```

**Other Implementation Details** Our reinforcement learning framework builds on verl Sheng et al. (2024), a flexible, efficient, and production-ready library for training large language models. By leveraging verl, we streamline our RL pipeline and maximize training throughput.

During RL, we generate images using a classifier-free guidance scale Ho & Salimans (2022) of 1.0. We found this is sufficient to generate meaningful images for the reward VLM model to grade and it can greatly accelerate the RL rollout speed as we don't need to generate the unconditioned images. During inference and evaluation, we use a classifier-free guidance scale of 5.0, the same as the default value of our base model Janus-Pro 7B.

## B EXPERIMENT DETAILS

### B.1 ANALYSIS ON TEXTUAL CoT

**Chain-of-Thought Word Frequency Analysis** Figure 5 exposes a clear pattern in REASONGEN-R1's chain-of-thought. First, it anchors each scene with high-level framing—"sense," "scene" and "natural" dominate, appearing in over 140% of CoTs—emphasizing overall context and realistic setting. Then, it refines visual style: terms like "soft," "highlights," "mood," and "sleek" (all above 100 %) specify lighting quality, emotional tone and texture.

Critically, the presence of "highlighting" and "emphasizing" (each in at least 70 % of CoTs) signals an explicit step to draw attention to the main subject. This reveals that REASONGEN-R1 doesn't merely describe objects; it actively plans compositional focus.

In addition to its core lexicon, REASONGEN-R1 draws on a sprawling array of less frequent modifiers—"background," to establish environmental context; "features," to spotlight salient visual elements; "calm," to evoke a serene atmosphere; "moments," to impart a sense of temporal capture; "captured," to underscore photographic realism; and many more—to infuse each reasoning sequence with subtle, context-specific nuance.

Overall, this analysis shows that REASONGEN-R1's chain-of-thought leverages complementary components—scene framing, style detailing, subject highlighting, and narrative enrichment—in concert to guide image generation.

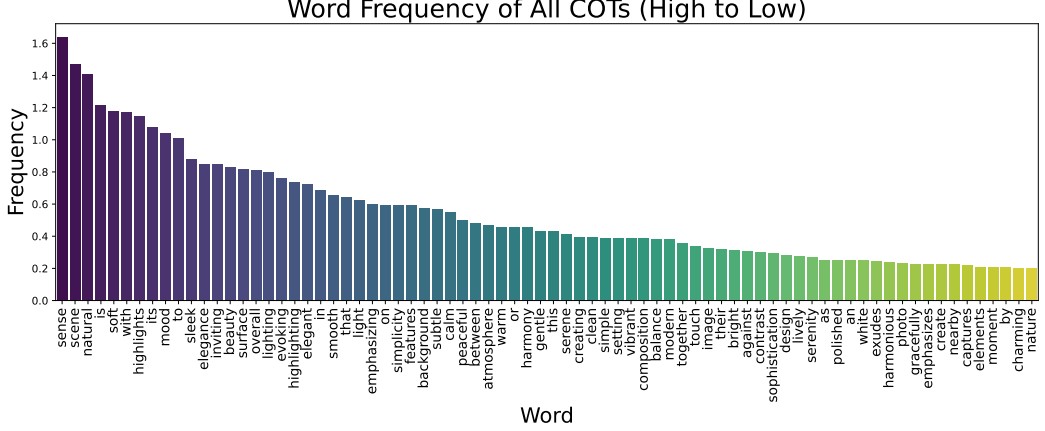

Figure 5: Ranked word-frequency distribution across 1000 Chain-of-Thought (CoT) rollouts. Only words appearing in at least 20% of CoTs are shown, and the three most common function words ("a", "an", and "and") have been removed to highlight more informative terms.

**Visual Sensitivity to Chain-of-Thought Substitutions** To further examine whether the chain-of-thought (CoT) genuinely guides the image generation process, we conduct a controlled substitution analysis. For each example, we selectively alter or add a specific element within the original CoT—such as object property, lighting condition, or background setting—while keeping the prompt and the rest of the reasoning unchanged. The goal is to isolate the impact of the modified token or phrase and observe how it propagates through the model's internal planning and ultimately manifests in the generated image.

As illustrated in Table 7, these targeted CoT substitutions lead to consistent and interpretable changes in the visual output. For instance, replacing "warm sunlight glows softly" with "bright sunlight shines" results in a stark contrast in overall lighting tone and shadow definition.

These results provide strong qualitative evidence that the model's generation is causally entangled with its reasoning process. The images do not merely correlate with the CoT—they reflect a coherent execution of its planning steps. This controlled substitution strategy thus offers compelling support for the claim that REASONGEN-R1 uses its CoT to explicitly anchor and shape each scene's content and style.

Table 7: More Visualization Results of REASONGEN-R1 with CoT Substitution

| Prompt | CoT | Image |
|---|---|---|
| A photo of a dog. | The **warm sunlight glows softly** on the elegant curves of this charming canine, highlighting her smooth fur and expressive eyes. The scene exudes a sense of peacefulness and joy, capturing the simplicity and beauty of a moment in nature. With a natural and re-laxed pose, the dog's relaxed yet alert posture speaks to her gentle curiosity and contentment. Against a softly blurred natural backdrop, this photograph highlights the natural elegance and grace of the scene, creating a timeless mood that celebrates the bond between humans and nature. | 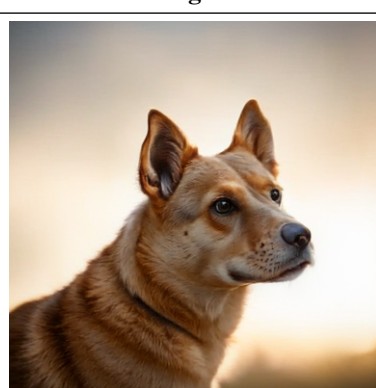 |
| A photo of a dog. | The **bright sunlight shines** on the elegant curves of this white husky, highlighting her smooth fur and ex-pressive eyes. The scene exudes a sense of peaceful-ness and joy, capturing the simplicity and beauty of a moment. With a natural and relaxed pose, the dog's relaxed yet alert posture speaks to her gentle curios-ity and contentment. Against a softly blurred natural backdrop, this photograph highlights the natural ele-gance and grace of the scene, creating a timeless mood that celebrates the bond between humans and nature. | 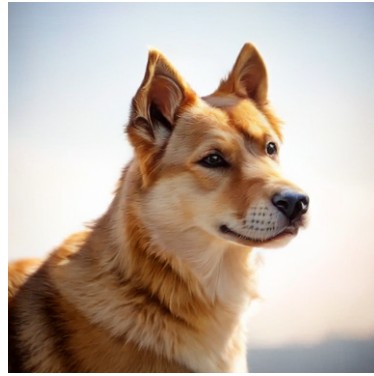 |
| A photo of a pot-ted plant left of a cow. | A serene scene features a cow peacefully standing in a **natural** setting, portraying a sense of peace and tran-quility. Beside the cow, a leafy green plant adds a touch of freshness to the image. With soft, natural lighting highlighting the cow's gentle features and the plant's vibrant color, the composition exudes a calm and har-monious atmosphere. The photo highlights the connec-tion between nature and animals. Perfect for showcas-ing the beauty of animals and nature, this image cap-tures a moment of balance and peace. | 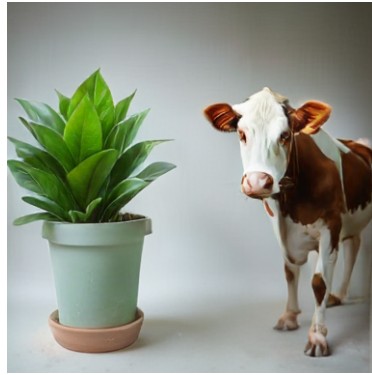 |
| A photo of a pot-ted plant left of a cow. | A serene scene features a cow peacefully standing in a **outdoor** setting, portraying a sense of peace and tran-quility. Beside the cow, a leafy green plant adds a touch of freshness to the image. With soft, natural lighting highlighting the cow's gentle features and the plant's vibrant color, the composition exudes a calm and har-monious atmosphere. The photo highlights the connec-tion between nature and animals. Perfect for showcas-ing the beauty of animals and nature, this image cap-tures a moment of balance and peace. | 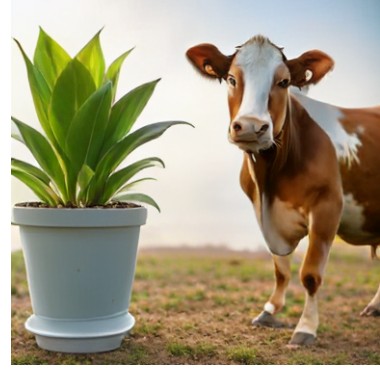 |

| Prompt | CoT | Image |
|--------|-----|-------|
| A photo of a bird. | A vibrant portrait of a stunning red-breasted cardinal perched gracefully against a soft, neutral background that highlights **the bird's rich plumage and lively expression**. Soft, natural lighting highlights the warm tones of the cardinal's bright red chest and contrasting black mask framed by a delicately shaped crest on its head. The composition captures **the bird's openness and gentleness**, evoking a sense of calm elegance and admiration for its majestic beauty. The overall mood is peaceful and uplifting, celebrating the elegance and majesty of this elegant harbinger of the season. | 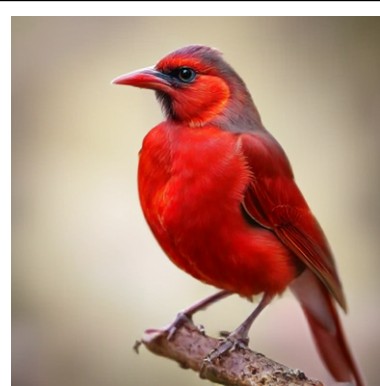 |
| A photo of a bird. | A vibrant portrait of a stunning red-breasted cardinal perched gracefully **beside a blooming wildflower**, set against a soft, neutral background that highlights **both the bird's rich plumage and the flower's delicate petals**. Soft, natural lighting brings out the warm tones of the cardinal's bright red chest and **the gentle hues of the flower**, while the bird's contrasting black mask and delicately shaped crest add refined detail. The composition balances the **cardinal's poised energy with the flower's fragile beauty**, evoking a sense of calm elegance and admiration for nature's fleeting wonders. The overall mood is peaceful and uplifting, celebrating the harmony and majesty of the season's natural treasures. | 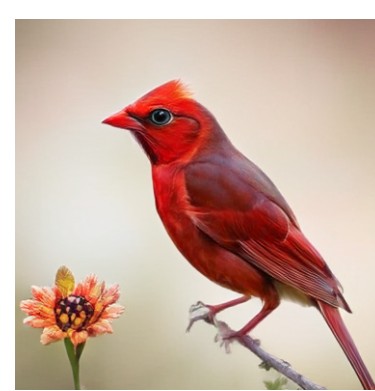 |
| A photo of a pizza. | A mouthwatering photo features a golden-brown pizza adorned with a variety of savory toppings, including **creamy cheese, plump Italian sausage slices, and juicy pepperoni**. The crispy crust edges around a deliciously flavorful filling, inviting all to savor the savory taste. Soft, natural light highlights the vibrant colors and textures of the ingredients, emphasizing their delicious appeal. Set against a clean background, the scene exudes a sense of freshness, warmth, and deliciousness, making the viewer crave a slice of this heavenly creation. The overall mood is inviting and appetizing, evoking a sense of satisfaction and joy. | 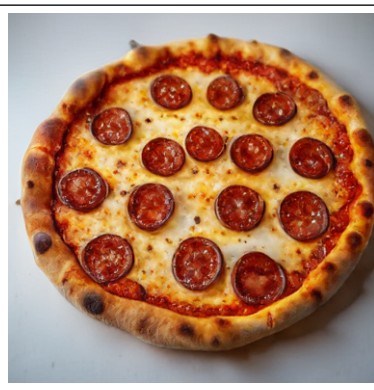 |
| A photo of a pizza. | A mouthwatering photo features a golden-brown pizza adorned with a variety of vibrant fruit toppings, including **sweet pineapple chunks and fresh tomatoes**. The crispy crust edges around a deliciously flavorful filling, inviting all to savor the unique fusion of sweet and savory. Soft, natural light highlights the bright colors and varied textures of the fruits, emphasizing their fresh and juicy appeal. Set against a clean background, the scene exudes a sense of freshness, warmth, and indulgence, making the viewer crave a slice of this heavenly creation. The overall mood is inviting and appetizing, evoking a sense of satisfaction and joy. | 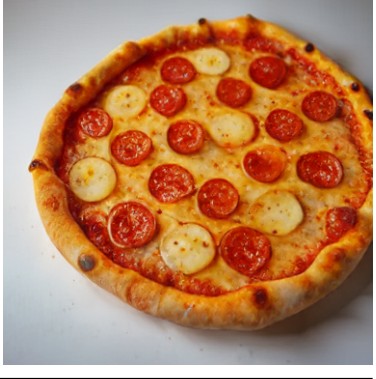 |

## B.2 EXPERIMENT SETTINGS

Table 8 summarizes the key settings used during supervised fine-tuning (SFT) and reinforcement learning (RL). These choices balance model capacity, sequence coverage, and compute efficiency for each stage of our pipeline.

Table 8: Hyperparameter Configurations

|  | SFT | RL |
| --- | --- | --- |
| Learning Rate | 1e-5 | 5e-6 |
| Optimizer | Adam | Adam |
| Weight Decay | 0.01 | 0.00 |
| KL Loss coeff | 0.00 | 0.00 |
| Rollout Batch Size | - | 32 |
| Rollouts per Prompt | - | 8 |
| Max Prompt Length | - | 156 |
| Max Response Length | - | 1024 |
| Max Sequence Length | 1792 | 1180 |
| Effective Batch Size | 128 | 16 |
| Steps | 1650 | 300 |
| Warmup Ratio | 0.03 | 0.00 |
| Epochs | 1 | - |
| GPU | 4 H100 80G | 4 H100 80G |
| Time to train | 8h | 32h |
| Total GPU Hours | 32 H100 | 128 H100 |
| CPU | Intel Xeon(R) Platinum 8480C | |

## B.3 CASE VISUALIZATION

In this section, we present visualizations of REASONGEN-R1's rollouts across four prompt categories: long and detailed prompts (Table 9), counting prompts (Table 10), spatial-relationship prompts (Table 11), and complex-attributes prompts (Table 12). The prompts in the first category are sampled from DPG-bench, while those in the remaining three categories are generated using Geneval.

Across all categories, the chain-of-thought (CoT) produced by REASONGEN-R1 aligns closely with the content of the generated images, demonstrating that its internal "thinking" effectively guides the planning and composition of each scene.

When using chain-of-thought (CoT) generation, REASONGEN-R1 consistently transforms terse prompts into richly detailed descriptions by specifying object attributes, lighting, textures, backgrounds, and overall mood. For instance, given the prompt "A photo of two persons" (Figure 10), REASONGEN-R1 first refines "two persons" into "a young woman and a man," then adds "natural light" and clothing details, follows with an ambient background description, and finally articulates the desired emotional tone of the scene.

Table 9: Visualization Results of REASONGEN-R1 on Detailed Prompts

| Prompt | CoT | Image |
|---|---|---|
| a tall giraffe standing next to a water hole, its long neck extended towards the lush leaves of an acacia tree. the tree's branches are just within reach of the giraffe's curious gaze. the scene is set in a sunlit savannah with the water hole reflecting the clear blue sky. | A serene scene unfolds under a bright blue sky, where sunlight illuminates a lush landscape with rolling hills and vibrant greenery. Tall, leafy trees frame the image, their broad canopies hinting at respite and shade. A majestic, graceful giraffe strikes a natural pose near the foreground, surveying the peaceful world above. Soft, warm light bathes the scene, highlighting the plant life and emphasizing the harmony between the giraffe and its natural surroundings. The mood is calm and tranquil, evoking a sense of wonder and peace as the viewer appreciates the beauty of this idyllic environment. The photograph captures a perfect balance of wildlife and natural beauty, highlighting the elegance and grace of the giraffe amidst a serene and lush setting. | 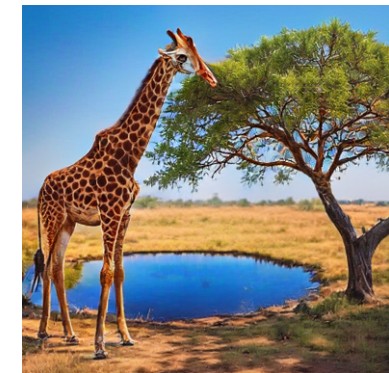 |
| A unique tree stands with its branches adorned with leaves that resemble vibrant purple balloons, glistening in the sunlight. The tree's trunk is a deep brown with a rough texture, contrasting sharply with the smooth, balloon-like foliage. Around the base of the tree, a bed of green grass provides a natural carpet, setting off the whimsical appearance of the tree's unusual leaves. | A breathtaking scene illuminates an enchanting landscape where a tree stands tall and proud, adorned with vibrant purple leaves that shimmer in the sunlight. Soft, gentle light highlights the delicate texture of the leaves and the subtle hues of the scene. Wisps of gentle breeze carry the sweet fragrance of fresh blossoms or lavender, creating a sense of serenity and peace. Nearby, a natural glow illuminates a soft, natural ground, with hints of green and brown framing the tranquil setting. This magical scene exudes a sense of wonder and harmony, evoking a feeling of pure delight and admiration for nature's beauty. | 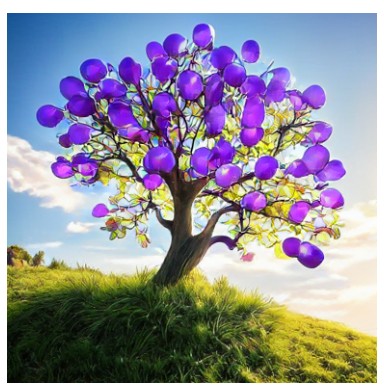 |

| **Prompt** | **CoT** | **Image** |
| --- | --- | --- |
| a patriotic-themed chopper motorcycle, its body emblazoned with the iconic red, white, and blue of the Stars and Stripes. The bike's gleaming chrome accents catch the light, highlighting its meticulous craftsmanship. Parked on a stretch of open road, the motorcycle's American flag motif stands out boldly against the asphalt. | A radiant scene features a sleek motorcycle parked on a quiet road, its brilliant chrome accents gleaming in the soft sunlight. The vibrant red and white elements of the bike contrast beautifully with the bold blue details, evoking a sense of freedom and adventure. Against a stunning sunset or sunset sky, the overall mood is serene and inspiring, highlighting the beauty of the open road and the magic of a summer evening. Together, the colors, lighting, and setting evoke a sense of freedom, celebration, and the endless possibilities of a journey. The overall mood is both peaceful and exhilarating, capturing the essence of a perfect summer evening. | 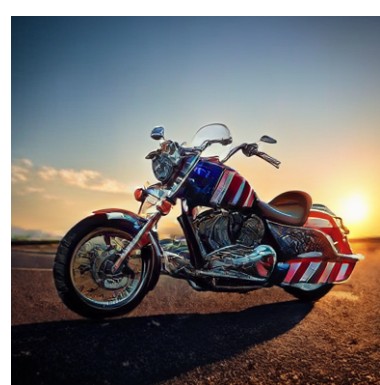 |
| a detailed oil painting that captures the essence of an elderly raccoon adorned with a distinguished black top hat. The raccoon's fur is depicted with textured, swirling strokes reminiscent of Van Gogh's signature style, and it clutches a bright red apple in its paws. The background swirls with vibrant colors, giving the impression of movement around the still figure of the raccoon. | A captivating scene features a wise and graceful figure with ornate attire, set against a lively, swirling background of rich, earthy tones. The artist's meticulous brushstrokes highlight the detailed textures of the figure's attire and the natural, textured backdrop, creating a harmonious contrast between elegance and elegance. The overall mood is one of mystery and calm, inviting viewers into a world where history and beauty entwine. Sharp, inviting eyes look out from the face, emanating wisdom and character, enhancing the captivating atmosphere of this timeless portrait. | 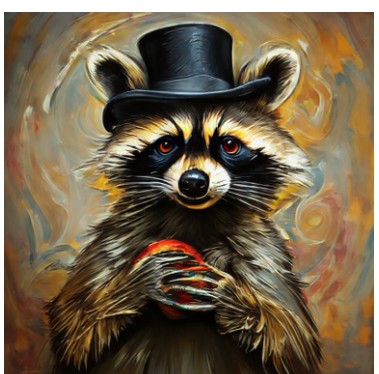 |

| Prompt | CoT | Image |
|--------|-----|-------|

An impressively detailed pencil illustration of Maggie Smith in the character of Reverend Mother is generating buzz on the ArtStation platform. The artwork, which has garnered awards for its lifelike quality, demonstrates a finesse reminiscent of Artgerm and Greg Rutkowski's dynamic strokes, with compositions that subtly hint at the influence of Alphonse Mucha's style. Its cinematic feel is accentuated by the careful play of light and shadow, earning acclaim and trending status among the art community.

The artwork depicts a serene and dignified figure wearing a traditional teaching or religious robe, with elegant details and a halo accentuating her spirit of wisdom and compassion. Taut lines and soft shading highlight the intricate details of her clothing and expression, evoking a sense of spiritual serenity and timelessness. Set against a subtle, monochromatic background that emphasizes simplicity and elegance, the image radiates a sense of gravitas and reverence. This striking piece receives accolades for its masterful lifelike portrayal, capturing the essence of a respected spiritual leader with grace and elegance.

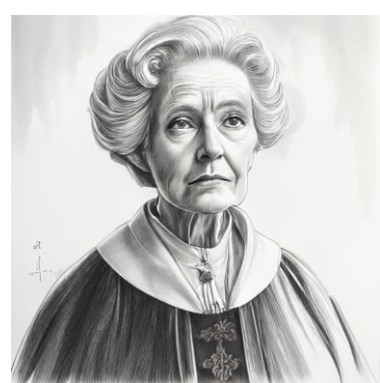

An ornate royal carriage, painted in deep red with golden trim, stands prominently against a landscape blanketed in pristine snow. Behind it, the silhouettes of tall pine trees dusted with white can be discerned through the soft haze of a winter's day. In front of the carriage, the snow-covered ground glistens under the subtle light of the afternoon sun.

A stunning scene captures a majestic red vehicle adorned with intricate golden accents, its elegance accentuated by the snowy landscape around it. Soft, golden sunlight bathes the scene in a warm glow, highlighting the glistening snow and the rich detailing of the vehicle. In the distance, a cluster of tall, snow-covered trees adds a serene and peaceful ambiance. Together, they evoke a sense of winter magic and serene beauty, evoking feelings of wonder and tranquility against the crisp winter backdrop.

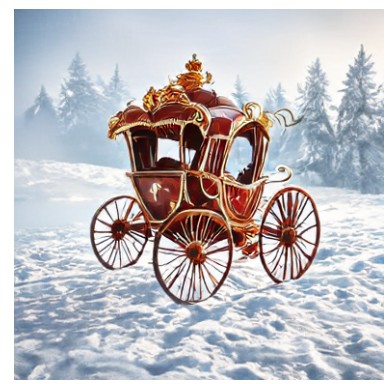

| **Prompt** | **CoT** | **Image** |
| --- | --- | --- |
| This is a vibrant, digitally-created watercolor illustration portraying an apocalyptic scene with sharp focus and a smooth finish. The artwork, by James Jean, features Rossdraws' signature style with elements reminiscent of Frank Frazetta's fantasy aesthetics, incorporating Mcbess's bold linework, and infused with the ethereal quality of Sakimichan's enchantments. The dynamic composition showcases a whirlwind of colors that vividly depicts the chaotic yet mesmerizing moment at the end of the world. | A vibrant and dynamic scene explodes with bursts of bright colors, showcasing a world in chaos. Skies blaze with fiery hues, contrasting beautifully with the glowing embers and sparks scattered across the landscape. Amidst swirling clouds of smoke and mist, eye-catching figures and creatures move between ruins and natural landscapes, emphasizing the aftermath of a towering catastrophe that left behind splashes of worry and hope. Amid the intensity, spontaneous bursts of light flickers across the scene, highlighting both destruction and resilience as the story continues. This digital masterpiece invigorates the senses with an electric energy, creating a powerful and emotive atmosphere that captures a world caught between ruin and renewal. | 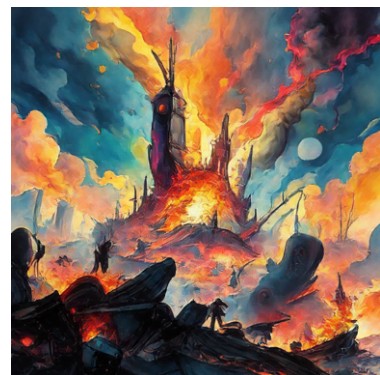 |
| A sleek, white laboratory designed with a blend of Matt Mahurin's moody aesthetic and Tsutomu Nihei's architectural sensibilities creates a stark, futuristic scene. The room features angular, geometric furniture with surfaces that have a smooth, matte finish, reflecting the dim, ambient lighting. Along the walls, various high-tech equipment and monitors display cryptic data, casting soft blue glows that contribute to the laboratory's enigmatic atmosphere. | The scene is illuminated by soft, bright lights that highlight the sleek and modern surfaces of the space. The air feels clean and crisp, with a subtle hint of technology and professionalism. A sense of calm and focus pervades the atmosphere, as a clean, orderly environment welcomes its inhabitants. The smooth, reflective surfaces emphasize the elegance of the surroundings, creating an inviting yet futuristic ambiance. Together, these elements evoke a sense of tranquility and purpose, as the space becomes a symbol of collaboration and innovation. | 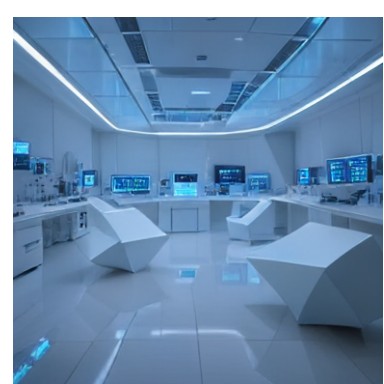 |

| Prompt | CoT | Image |
|---|---|---|

A digitally rendered image of the iconic Monalisa capturing a selfie, boasting 8K resolution and hyper-realistic details that highlight the delicate textures of her skin and the intricate fibers of her clothing. The scene is illuminated with cinematic lighting that casts soft shadows and enhances the depth of field, giving a three-dimensional quality to the image. The background is blurred artfully, drawing full attention to her enigmatic expression and the modern device in her hands, all created with the precision of an Octane render engine.

Soft, natural light illuminates a warm, inviting scene where a person stands confidently before a breathtaking landscape. Behind them, the rolling hills and luscious greenery create a serene, peaceful backdrop. The individual wears a flowing dress adorned with intricate patterns and sheer sleeves, exuding elegance and grace. The gentle curve of the horizon and the subtle glow on the landscape hint at the early morning or late afternoon glow of the setting sun. Together with subtle lighting emphasizing the textures of the fabric and the natural environment, the overall mood is warm, tranquil, and harmonious, evoking a sense of awe and appreciation for nature's beauty. The artistry of a dream-like atmosphere adds to the timeless and refined mood of the scene.

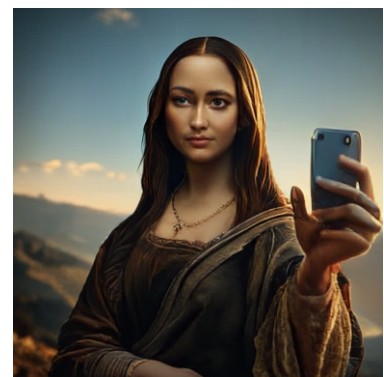

a digital illustration of an adorable baby penguin emoji, sporting a vibrant blue hat on its head and snug red gloves on its flippers. The penguin is dressed in a bright green shirt that contrasts with its sleek black and white feathers, and it's wearing cheerful yellow pants that add a pop of color. The emoji is set against a clean, white background, making the colorful attire of the penguin stand out even more.

A charming scene features a lively and stylish character wearing a bright, cheerful hat and a cheerful expression. The character sits comfortably, with feet touching the surface, and is adorned with simple but elegant accessories that add a touch of elegance. The soft, clean background highlights the character's playful yet sophisticated appearance, emphasizing a sense of warmth and joy. The setting is simple and minimalistic, allowing the character to shine and evoke feelings of happiness and calmness. Overall, the image conveys a sense of fun, elegance, and happiness.

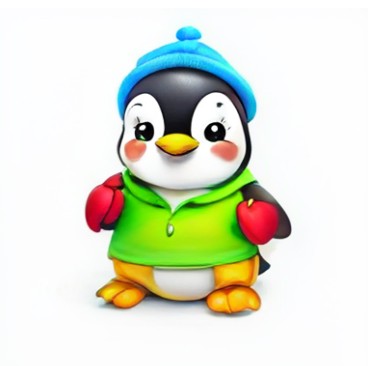

| **Prompt** | **CoT** | **Image** |
|---|---|---|
| A close-up image of an intricately designed lotus flower, which appears to be crafted entirely from crystal-clear water droplets. The flower is set against a backdrop of soft green lily pads floating on a tranquil pond. Sunlight filters through the scene, highlighting the delicate texture and the shimmering surface of the water-formed petals. | A serene scene unfolds with sunlight filtering through the leaves, highlighting the gentle curves and clear, crystalline surfaces of the graceful flowers. The vibrant green pond reflects the sky above, and a soft breeze caresses the leaves, creating a sense of calm and freshness. As the focus shifts to a close-up of a single flower, its delicate petals are captured under a soft, natural light that emphasizes their elegance and transparency. The flowers bloom slowly open, revealing their intricate structure and subtle colors beneath the water surface. The overall mood exudes peace, harmony, and a sense of delicate beauty, evoking a feeling of quiet tranquility and appreciation for the natural world. | 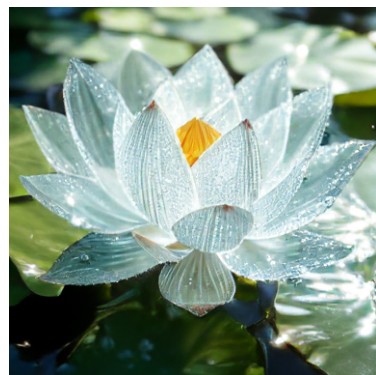 |
| In the art piece, a realistically depicted young girl with flowing blonde hair gazes intently into the distance, her eyes reflecting the vibrant hues of a spring forest. The verdant greens and soft pastels of the budding trees are captured in subtle brushstrokes, giving the scene a serene and tranquil atmosphere. The minimalist composition focuses on the girl's expression of wonder and the lush woodland background, while the texture of the oil paint adds depth and richness to the canvas. | A tender scene features a young girl standing gracefully amidst a blossoming world of flowers and trees, with sunlight filtering softly through the leaves overhead. Her expression is one of serene wonder, as if she's caught in the moment by the magic of spring. Soft pastel hues of green, white, and pink are the backdrop to this enchanting setting, evoking a gentle, peaceful mood. The gentle petals of the nearby flowers and the soft texture of the trees create a sense of harmony and transience, inviting the viewer to share in the simple beauty of this fleeting springtime moment. | 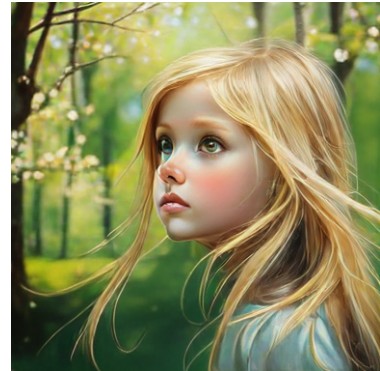 |

| **Prompt** | **CoT** | **Image** |
|---|---|---|
| a view through a window pane speckled with raindrops, showcasing a cityscape of tall buildings with reflective glass facades. the gray overcast sky looms above the urban skyline, and the raindrops create a blurred effect on the structures in the distance. the window's frame is a stark white, contrasting with the muted colors of the city beyond. | A subtle, atmospheric scene unfolds with a modern, light-colored window frame that frames a crisp view of a city shaped by towering structures and bustling urban life. Even through the hazy, overcast sky, you can make out textural details etched into the glass, highlighting the city's vibrant energy and hinting at moisture on the surface. Soft, diffused light highlights the moody and serene atmosphere, evoking a sense of calm introspection. The gentle highlights accentuate the city's architectural elements, as delicate raindrops fall like soft tracers across the image, adding a sense of ephemeral beauty to this captivating scene. Together, the composition and mood combine to evoke a moody, yet uplifting mood, like a fleeting moment of peace in the midst of urban life. | 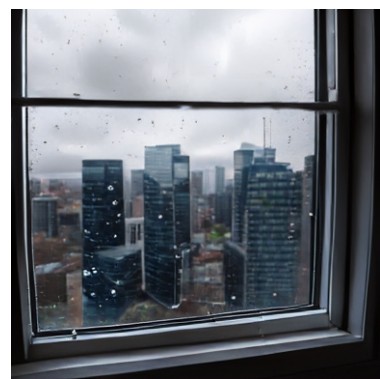 |
| a peculiar sight of a tree with vibrant yellow leaves, each leaf delicately edged with hints of autumnal orange. Among the branches hang unusual blue apples, their smooth surfaces reflecting the soft sunlight. The tree stands alone in a field, its roots sprawling across the rich, brown earth. | A close-up scene of a tree adorned with vibrant yellow leaves perched atop a warm brown tree trunk, with soft, natural lighting that highlights the intricate details of the leaves and the bark. The sunlight gently illuminates the bright yellow hue of the leaves and highlights the fresh orange tips that edge them. A few delicate leaves have softly embraced the branches, adding a sense of calm and beauty to the setting. Along the tree, several hanging eggs of blue, perfectly in sync with the natural light, bring an air of mystery and harmony. The scene exudes a peaceful, quiet mood, emphasizing the natural beauty of the tree and its surroundings, evoking a sense of calm wonder as the gentle balance of yellow, orange, blue, and green colors blend into a harmonious visual experience. | 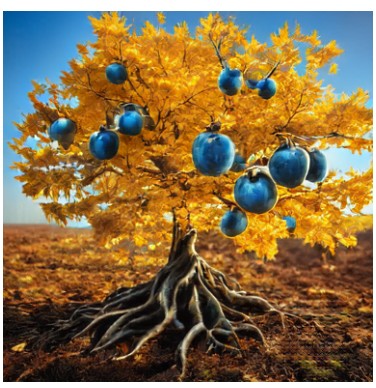 |

*continued from previous page*

| Prompt | CoT | Image |
|--------|-----|-------|

a striking portrait that captures the essence of Salvador Dalí, with one side of his face depicted in his iconic, surrealistic style, and the other half transformed into a metallic, robotic visage. The painting features a vivid array of colors, with the robotic side incorporating shades of silver and hints of circuitry, contrasting with the warm, flesh tones of Dalí's human side. The background is a simple, solid color to ensure the focus remains on the intricate details of Dalí's dual representation.

A captivating image features a person with an intriguing blend of realistic and stylized elements. Their face showcases a serene and contemplative expression, highlighted by vibrant colors, rich textures, and bold contrasts. The background is soft, subtly complementing the subject's artistic appearance and evoking a sense of mystery and wonder. The lighting accentuates the details of the face and highlights the interplay of smooth and shimmering surfaces, capturing a moment of elegance and imagination. Together, these elements convey a mood that is both alluring and thought-provoking, inviting viewers to explore the beauty and complexity of the artwork.

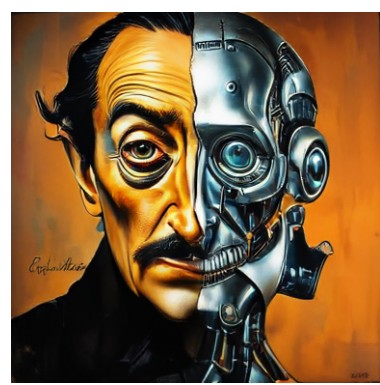

An aerial view of Toronto's skyline dominated by the iconic CN Tower standing tall amongst the surrounding buildings. The image is taken from the window of an airplane, providing a clear, bird's-eye perspective of the urban landscape. Across the image, the words "The CN Tower" are prominently displayed in the playful Comic Sans font. The cluster of city structures is neatly bisected by the glistening blue ribbon of a river.

A stunning view of a vibrant city under a bright, sunny sky showcases iconic buildings, bustling streets, and a lively atmosphere. As the camera moves into the background, the panoramic scene unfolds a harmonious blend of nature and architecture, highlighting the urban splendor. Amidst this lively setting, a sleek helicopter or plane initially appears in the mid-ground, suggesting a sense of perspective and movement. Soft, warm lighting enhances the overall mood, evoking feelings of hope, harmony, and curiosity. Amidst subtle hints of bustling activity and cheerful colors, the focus subtly shifts to radiant natural lighting illuminating the scene. The scene effortlessly highlights the vibrant energy of the city, its impressive architecture, and the simple beauty of the sky and water, all captured within a captivating moment filled with wonder.

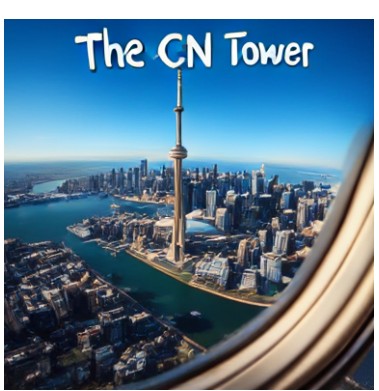

| Prompt | CoT | Image |
|---|---|---|
| A vibrant graffiti artwork displaying the word "WOMBAT" in bold, multicolored letters, each character outlined in black to create a striking contrast against the stark white wall. The letters are embellished with various shades of blue, green, red, and yellow, with dramatic splashes of paint scattered around the composition. The texture of the dripping paint adds a dynamic and tactile quality to the mural. | A lively scene features a pristine white wall adorned with a striking artistic display of vibrant patterns and colors. Bright, bold brushstrokes highlight the cheerful, dynamic energy of each page-turning page of the comic, evoking an inviting and playful atmosphere. Soft, natural lighting highlights the smooth texture of the surface, emphasizing the contrast between the clean background and the vivid designs. Tiny splotches of paint peeking through the edges add a sense of spontaneity, evocative of the creativity that this space now holds. Together, the elements of this scene create a sense of mystery and delight, inviting viewers into a world full of wonder and artistic expression. The viewing experience is both energizing and peaceful, celebrating creativity and color. | 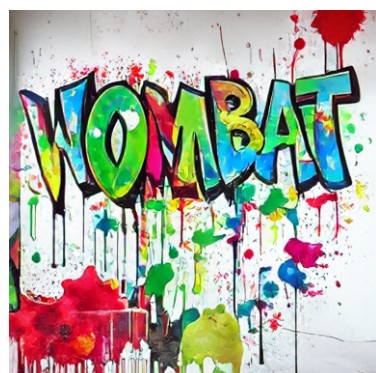 |

Table 10: Visualization Results of REASONGEN-R1 on Counting Prompts

| Prompt | CoT | Image |
|---|---|---|
| A photo of two persons. | A scene features a young woman and a man standing together, dressed in elegant outfits that complement their sophisticated and stylish appearance. The natural lighting highlights their kind expressions and highlights the fine details of the fabrics and textures in their clothing. The background adds a subtle and stylish ambiance to the scene, emphasizing the pair's poise and elegance. The overall mood is warm and harmonious, exuding a sense of sophistication and elegance. | 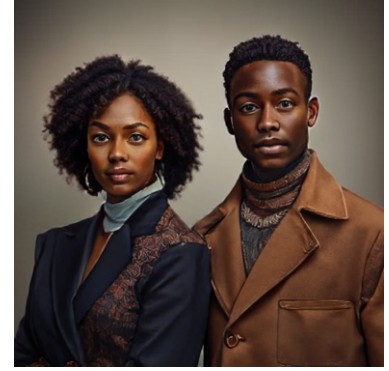 |

*continued from previous page*

| Prompt | CoT | Image |
|---|---|---|
| A photo of two donuts. | A beautifully presented photo showcases a plump, lightly glazed donut resting on a clean surface, highlighting its soft, airy texture and inviting appearance. The background is softly lit, emphasizing a clean and simple setting that focuses attention on the donut. Soft, natural lighting highlights the gentle sheen of the glaze, highlighting its smooth surface and subtle curves. The overall mood is warm and inviting, evoking a sense of freshness and simplicity. The composition emphasizes the luscious detail of the donut's surface, while the background remains unobtrusive, allowing the viewer to appreciate the natural beauty of this delightful treat. |  |
| A photo of three vases. | A striking scene features three slender, elegant vases carefully arranged on a pristine surface, set against a softly blurred background that highlights their graceful forms. Soft, natural lighting emphasizes the glossy surface of the vases, highlighting the smooth curves and subtle sparkle of the glass. The composition exudes a sense of calm and sophistication, highlighting the harmony and simplicity of the arrangement. Together, these vases create a sense of balance and serenity, evoking a feeling of stillness and beauty. The overall mood is peaceful and elegant, inviting viewers to appreciate the harmony between form, light, and space. |  |
| A photo of two cakes. | A beautifully crafted photo features a pristine, elegant white cake adorned with delicate frosting and subtly sparkling accents. Soft, natural lighting highlights the texture of the smooth icing and the detailed decorations, evoking a sense of sophistication and celebration. Beside the cake, a similarly pristine pastel-colored cake adds a touch of color and contrast, harmonizing with the overall refined and joyful mood. Together, the two cakes create a balanced and harmonious composition, presenting a scene of celebration and sweetness. The scene is set against a simple, neutral background that underscores the cakes as the focal point, emphasizing their elegance and charm. |  |

*continued from previous page*

| Prompt | CoT | Image |
|---|---|---|
| A photo of three birds. | The photo depicts three vibrant and lively birds perched together, set against a softly muted background that highlights their striking colors. The scene exudes a sense of harmony and natural beauty, showcasing each bird's unique features and colors. Soft, natural lighting illuminates their smooth feathers, emphasizing their smooth texture and the gentle contours of their bodies. The overall mood is calm and peaceful, evoking a sense of serenity. The composition emphasizes balance and balance, with the birds evenly spaced to highlight their individual charm and togetherness. This image captures the essence of these colorful birds in a way that emphasizes their natural beauty and the tranquility of their environment. | 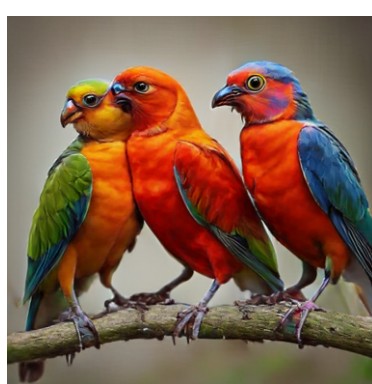 |
| A photo of three cows. | A warm and serene scene captures a trio of brown cows standing close together. Soft natural light highlights their calm expressions and highlights the natural beauty of their smooth coats and gentle eyes. The background hints at a rural setting, emphasizing a peaceful and pastoral atmosphere. The overall mood exudes tranquility and a sense of humble contentment, evoking feelings of calm and serenity. | 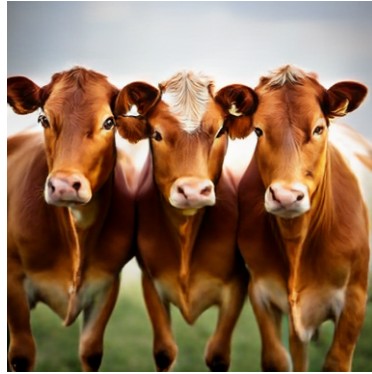 |
| A photo of two refrigerators. | A sleek and modern image showcases two clean white refrigerators standing side by side, highlighting their simple and elegant design. Soft, natural lighting highlights the smooth surfaces and highlights the crisp lines of the units. The minimalist background ensures that the focus remains on the fresh and crisp look of the refrigerators. The overall mood is calm and serene, emphasizing cleanliness and simplicity. The composition emphasizes balance and harmony, evoking a sense of order and everyday convenience. The photo highlights the timeless beauty of these household appliances in their everyday surroundings. | 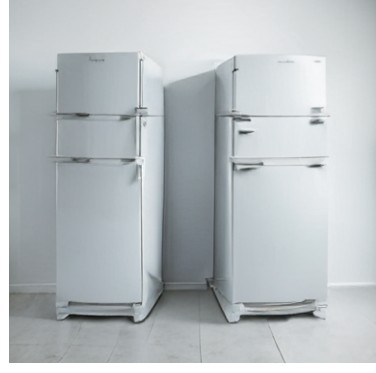 |

*continued from previous page*

| Prompt | CoT | Image |
|---|---|---|
| A photo of two beds. | A serene and elegant scene features a restful room with soft, ambient lighting highlighting a plush bed draped in crisp, white linens. The bed's simple yet sophisticated design showcases a smooth headboard and other subtle details, creating a sense of calm and relaxation. Soft, natural light filters in, highlighting the inviting atmosphere. A decorative bedside table with modest decor completes the scene, evoking a peaceful and welcoming mood. | 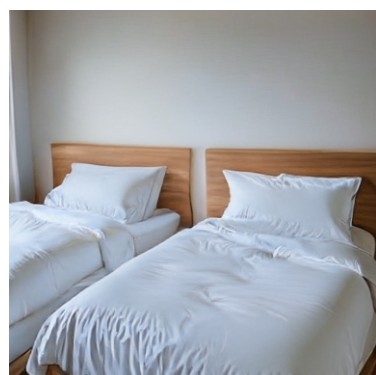 |
| A photo of three teddy bears. | A charming and cozy scene features three adorable teddy bears sitting close together, each exuding a sense of warmth and friendship. Soft, natural lighting highlights their plush textures and warm, inviting tones. The lighting casts gentle highlights on their fur, accentuating their cuddly qualities. The atmosphere is serene and peaceful, evoking a sense of comfort and tranquility. The composition captures a harmonious balance between the bears, emphasizing their companionship. Overall, this photo exudes sweetness and a feeling of joy, highlighting the simple pleasures of cuddling with these stuffed friends. | 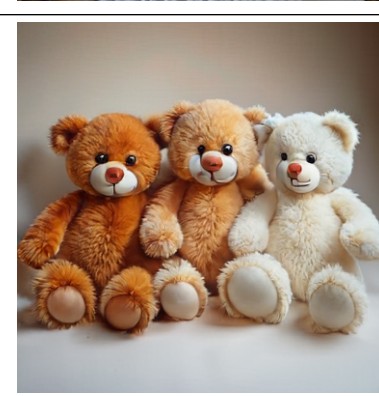 |
| A photo of two apples. | This cozy image captures a single, beautifully smooth apple resting on a smooth surface. The apple's bright, fresh red hue contrasts softly against the serene backdrop. Gentle lighting highlights its natural sheen and subtle curves, accentuating its plump round shape. The overall mood exudes a sense of freshness, health, and simplicity, with soft, natural lighting highlighting the apple's inviting glow. The mood is calm and natural, emphasizing the apple's organic appeal. The composition is simple yet striking, focusing solely on the apple to evoke feelings of comfort and cleanliness. Natural, soft lighting enhances the subject, evoking a sense of freshness and health. Overall, the image creates an atmosphere that celebrates the simple beauty and purity of nature. | 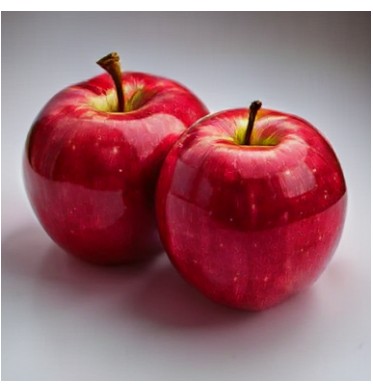 |

| Prompt | CoT | Image |
|---|---|---|
| A photo of four potted plants. | A vibrant scene features a simple yet elegant composition with a clean background highlighting a small but lively grouping of young plants and a young sapling. Soft, natural lighting highlights the fresh green hues and subtle details of each plant, emphasizing their lively freshness and growth. Their natural, naturalistic arrangement emphasizes harmony and balance, evoking a sense of calm and serenity. Together, they create a feeling of new beginnings and natural beauty, with gentle shadows adding depth and a touch of softness to the scene. This serene and peaceful composition captures the essence of growth and renewal, celebrating the simple elegance of nature. | |
| A photo of three traffic lights. | A trio of sleek, modern traffic lights stands confidently against a serene background, each signaling to approaching drivers or pedestrians a calm and organized passage. The warm glow of the lights illuminates their sturdy metal frames, highlighting their functional design and radiating a sense of urban safety. Together, they create a balanced and orderly scene, evoking a feeling of orderly movement and community. The natural lighting accentuates their purpose as vital transportation tools, seamlessly blending style and functionality. | |
| A photo of two spoons. | A sleek, elegant photo features a single, polished silver spoon and a matching spoon positioned side by side. The surface of the spoons reflects soft, natural lighting, highlighting their smooth and shiny surfaces. The background is simple and neutral, emphasizing the elegance of the objects. Subtle highlights create a sense of depth, drawing attention to the craftsmanship and quality of the spoons. Overall, the image exudes a sense of sophistication and purity, emphasizing the beauty and serenity of the scene. | 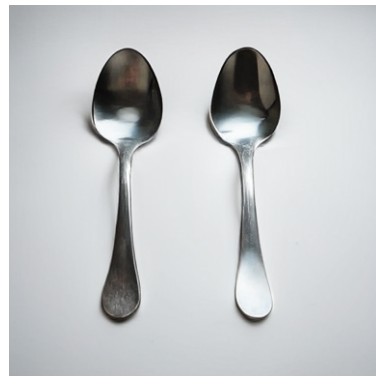 |

*continued from previous page*

| Prompt | CoT | Image |
|---|---|---|
| A photo of three cats. | A beautifully composed scene features a sleek, graceful cat with soft, shimmering fur in a relaxed pose, alongside a lively and curious kitten with sparkling eyes and playful ears, and a lively, animated cat with an alert expression and bright, expressive eyes. Together, they create a charming and lively mood, highlighting their natural beauty, individual personalities, and the joy of companionship. The lighting highlights the smooth texture of their coats and highlights each of their unique expressions, evoking a sense of warmth, playfulness, and harmony. This photo captures a timeless moment of connection and friendship, inviting a viewer to feel a sense of peace and contentment. The overall mood is heartwarming and enchanting, showcasing these little wonders in their full glory. | 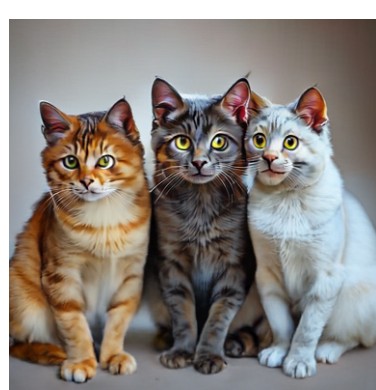 |
| A photo of four zebras. | A sleek row of horses stands gracefully, each adorned in elegant black-and-white stripes that shimmer under soft, natural light. Their flowing manes showcase a beautiful harmony of shades, enhancing their refined presence. The horse on the left presents a confident and poised stance, while the others stand tall and serene. Together, they evoke a sense of harmony and grace, celebrating the beauty of each individual's unique pattern against a peaceful, softly lit background. The overall mood is calm and elegant, evoking a sense of serenity and elegance. | 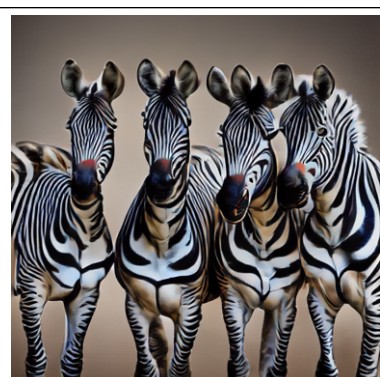 |

Table 11: Visualization Results of REASONGEN-R1 on Spatial Relationship Prompts

| Prompt | CoT | Image |
|---|---|---|
| A photo of a stop sign below an orange. | A warm, inviting scene features a vibrant orange resting gently on a surface, highlighting its bumpy texture and natural curves. Nearby, a classic red stop sign stands upright, creating a beautiful contrast with the orange's bright hue. Soft, natural light illuminates the scene, highlighting the glossy surface of the orange and the crisp edges of the sign. Together, they evoke a sense of balance and harmony, with the orange providing a pop of color that complements the boldness of the stop sign. The overall mood is calm and inviting, highlighting the connection between these two simple yet striking elements. | 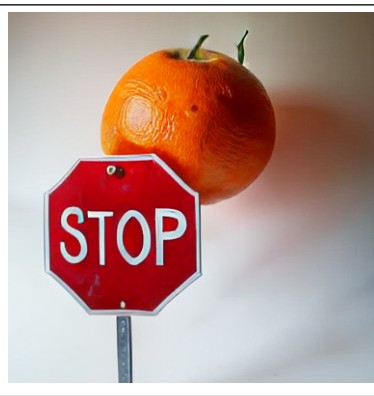 |

*continued from previous page*

| Prompt | CoT | Image |
|---|---|---|
| A photo of a sheep left of a bird. | A soft and peaceful scene features a gentle and charming white-coated sheep standing serenely with a calm demeanor, highlighting its peaceful presence. Nearby, a graceful bird with lively feathers perches calmly, adding a touch of serenity to the setting. Together, they create a harmonious and tranquil atmosphere, evoking a sense of nature's serenity and harmony. Soft, natural light highlights their peaceful expressions and the subtle details of their feathers and fur, making this composition both soothing and exquisite. The overall mood is peaceful and calming, inviting viewers to appreciate the quiet beauty of this natural moment. | 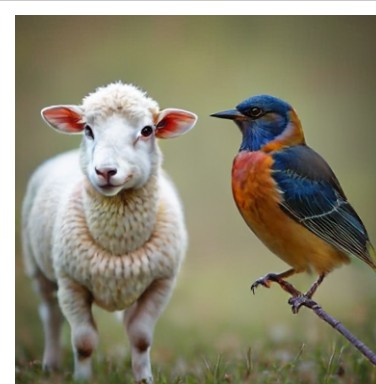 |
| A photo of a cup left of a vase. | A elegant and sophisticated scene features a sleek, stylish vase next to a clean and crisp cup. The setting is simple, highlighting the beauty of the objects and their harmonious pairing. Soft, natural lighting highlights the smooth surfaces and delicate curves of both the vase and cup, creating a sense of calm and balance. Together, they evoke a sense of simplicity, elegance, and harmony, emphasizing the artfulness and elegance of the composition. The overall mood is serene and elegant, inviting the viewer to appreciate the beauty of these simple yet refined objects. | 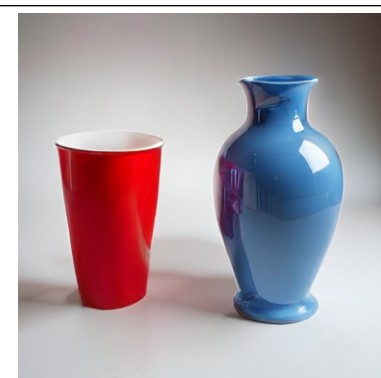 |
| A photo of a potted plant left of a cow. | A serene scene features a cow peacefully standing in a natural setting, portraying a sense of peace and tranquility. Beside the cow, a leafy green plant adds a touch of freshness to the image. With soft, natural lighting highlighting the cow's gentle features and the plant's vibrant color, the composition exudes a calm and harmonious atmosphere. The photo highlights the connection between nature and animals. Perfect for showcasing the beauty of animals and nature, this image captures a moment of balance and peace. | 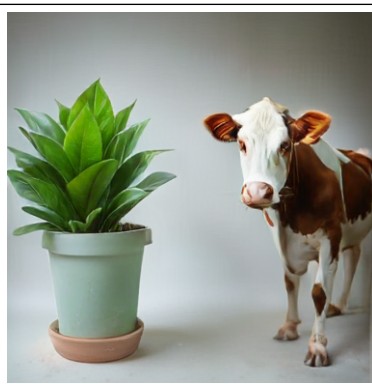 |
| A photo of a bench left of a sandwich. | A charming scene features a sleek, inviting wooden bench set against a natural backdrop of greenery and soft sunlight. A smooth, welcoming path leads up to the bench, where you can imagine relaxing or spending peaceful moments. Nearby, a delicious-looking sandwich crafted with fresh ingredients sits on a rustic wooden surface, highlighting its wholesome appeal. Together, these elements create a sense of tranquility and simplicity, evoking feelings of calm and satisfaction. The overall mood is warm and inviting, emphasizing the beauty of nature and the joy of savoring a good meal. This picturesque scene captures a perfect moment of relaxation and contentment. | 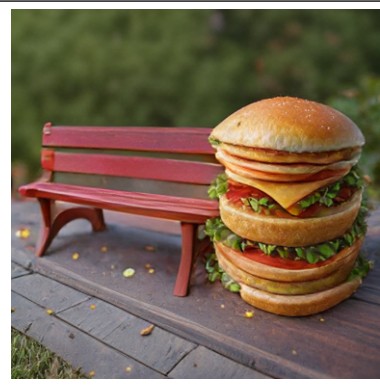 |

*continued from previous page*

| Prompt | CoT | Image |
|---|---|---|
| A photo of a bowl above a couch. | A sleek and elegant photo features a stylish off-white or light gray sofa with a simple, clean design, showcasing its inviting surface and smooth leather or fabric texture. Against a soft, neutral background, the sofa provides a simple yet sophisticated setting that highlights its timeless craftsmanship. On the surface of the sofa, a luxurious bright cushion or some casual seating accessories accentuate the scene's balance and highlight the contrast between relaxation and elegance. A single spotlight illuminates the scene, emphasizing the smooth surfaces, the subtle sheen of the sofa, and the crisp texture of the cushions, evoking a sense of calm and sophistication. The overall mood is warm and inviting, capturing the perfect atmosphere of relaxation without being too flashy. High-quality lighting and composition highlight the details of the sofa and the accessories, while the thoughtful arrangement creates a sense of balance and harmony. The overall mood is luxurious and serene, inviting viewers to imagine themselves in a comfortable and elegant space. | 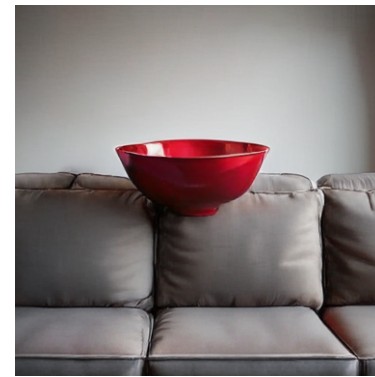 |
| A photo of a bench below a cake. | A charming scene features a compact and elegant wooden bench positioned beside a sparkling cake, both exuding a sense of celebration and joy. The cake stands tall with carefully decorated layers and vibrant icing, its light frosting gleaming under soft, inviting lighting. The bench is simple yet attractive, showcasing its natural wood grain and craftsmanship, inviting you to sit and enjoy this moment of togetherness and celebration. Together, they create a harmonious composition that captures a sense of warmth and happiness, perfect for a special occasion or simply sharing a delicious treat. The overall mood is bright, uplifting, and festive, evoking feelings of joy and celebration. | 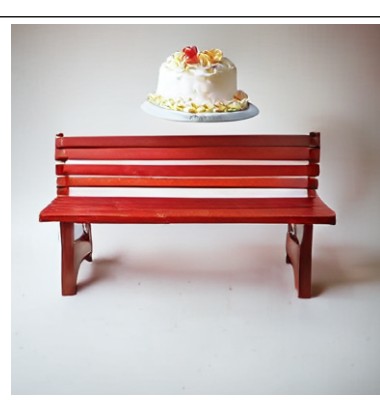 |
| A photo of a parking meter left of a fire hydrant. | A sleek and shiny photo captures a streamlined and reflective metal surface of a modern lamp post, gracefully standing tall against a softly blurred city background. Nearby, a charming fire hydrant with a crisp red finish and gleaming gold accents catches the eye with its elegant curves, adding a touch of classic charm. Together, these elements evoke a sense of urban elegance and timeless style, highlighting the harmony between modern and traditional design. Soft, natural lighting highlights their polished surfaces and highlights, evoking a calm, inviting atmosphere under a clear sky. This scene blends the beauty of urban architecture with the warm glow of sunshine, creating an image that exudes both sophistication and serenity. | 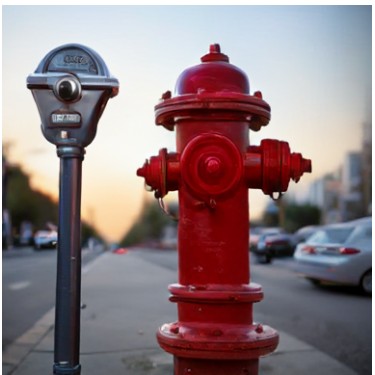 |

*continued from previous page*

| Prompt | CoT | Image |
|--------|-----|-------|
| A photo of a motorcycle right of a tv. | A sleek black motorcycle sits beside its elegant black screen, its sharp lines and powerful presence exuding sophistication and modernity. Soft, natural light highlights the smooth surfaces and gleaming chrome accents, creating a sense of elegance and purity. The mood is calm and captivating, emphasizing the perfect balance between beauty and technology. This scene is both timeless and contemporary, blending the best of the past and the future into a harmonious and stylish showcase of craftsmanship and design. | 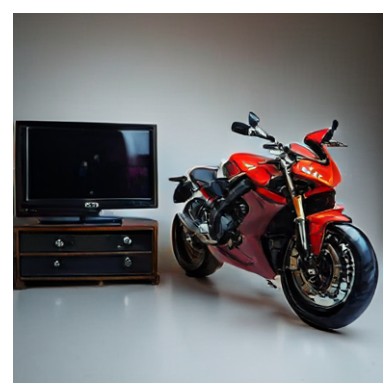 |
| A photo of a bear left of a donut. | A charming and majestic bear stands peacefully in a natural setting, exuding a sense of calm serenity. Soft, warm lighting highlights its rich fur and highlights the natural beauty of the scene. The bear's gentle expression is accentuated by the gentle lighting that creates a peaceful mood. Close to the bear's side, a colorful, artistic depiction of a sweet donut appears, with elaborate details highlighting its shape and toppings. The contrast between the natural, wild setting and the fun, playful donut creates a delightful juxtaposition, evoking a feeling of harmony and joy. This delightful scene captures the balance between the beauty of nature and the sweet, simple pleasures of donuts, making it a heartwarming and inviting image. | 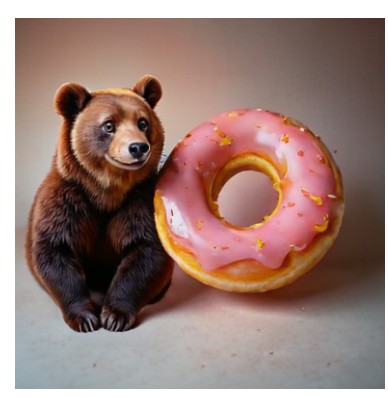 |
| A photo of a dog right of a tennis racket. | A sleek, shining black and white tennis racket rests gracefully on the ground, its strings forming a delicate pattern. The scene is bathed in soft, natural light that highlights the smooth surface of the racket and emphasizes its elegance and precision. Nearby, a lively and playful dog sits upright, gazing curiously at the racket with bright, alert eyes. Together, they create a harmonious and balanced composition that evokes feelings of curiosity, sportsmanship, and an appreciation for the beauty of simple, everyday moments. The photograph captures the moment, where the racket and the dog bask in a peaceful, sunlit setting that blends outdoors and the allure of a bright, welcoming day. The soft focus and balanced lighting evoke a sense of calm, highlighting the connection between the natural elements and the sense of harmony and balance in this bright and captivating scene. | 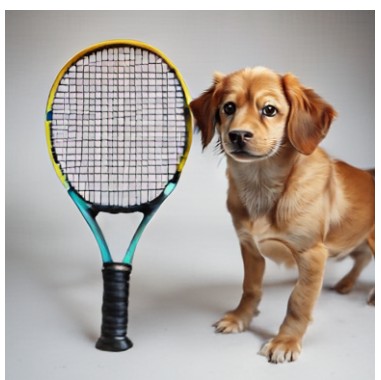 |

*continued from previous page*

| Prompt | CoT | Image |
|--------|-----|-------|
| A photo of a potted plant left of a bird. | A delicate scene features a lively green plant gently standing tall, its leaves reaching upwards with a sense of grace and elegance. Next to it, a graceful bird perches calmly, its eyes watching the world with quiet curiosity and awareness. Together, the plant and bird create a serene, peaceful mood that highlights the beauty and harmony of nature. Soft, natural lighting brings out the subtle details of their surroundings, enhancing the sense of calmness and tranquility in this lovely composition. The overall mood is one of balance, wonder, and appreciation for the gentle co-existence of plant and bird in their natural habitat. | 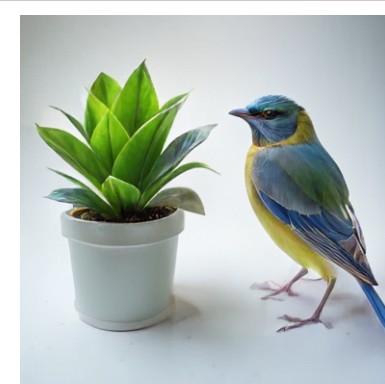 |
| A photo of a bottle right of a donut. | A beautifully simple yet appetizing scene features a vibrant red jelly-filled doughnut resting on a classic white plate, highlighting its delicious, soft texture and moist filling. To the side, a charming glass of cold, refreshing soda adds a refreshing contrast, with the sparkle of bubbles and ice cubes dancing in the light. The lighting highlights the fresh, appetizing colors and creates a sense of calmness and simplicity. Together, they evoke a sense of warmth, sweetness, and a lovely moment of indulgence. The overall mood is inviting and joyful, evoking the feeling of enjoying a lovely snack in a bright, sunlit setting. | 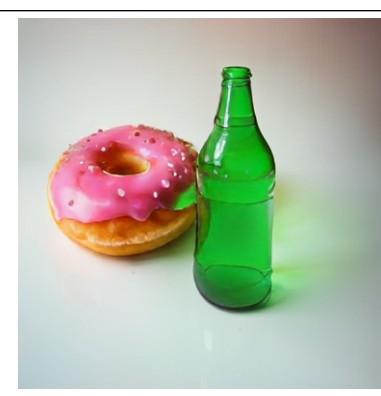 |
| A photo of a cell phone right of a bottle. | A sleek image features a beautifully crafted green bottle with a tall, slender neck and a subtle curve along its body. The bottle sits calmly, showcasing its elegant lines and refreshing, natural appearance. Nearby, a sleek smartphone displays a simple, clean screen that highlights the minimalist design of its surface and the sleek edges of its body. Together, the image emphasizes harmony and balance, presenting a scene of natural beauty and simplicity. Both objects are placed softly in natural, soft lighting, which highlights their polished surfaces and evokes a sense of freshness and tranquility. The overall mood is peaceful and serene, accentuating the purity and elegance of both the bottle and the smartphone. The composition draws the viewer's eye to the contrast between the two objects, celebrating a perfect blend of elegance and simplicity. | 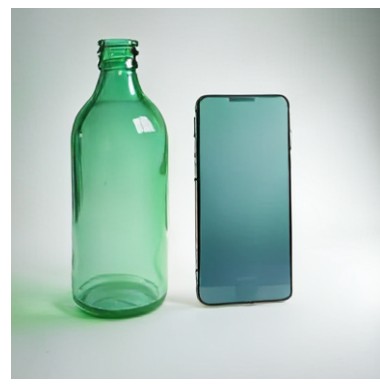 |

| Prompt | CoT | Image |
|---|---|---|
| A photo of a cake above a boat. | A serene scene features a clean, elegant white or pastel-colored boat gently resting on calm, reflective waters. On the boat's surface, a delicate, well-crafted cake sits resplendent, inviting admiration. Soft, natural lighting highlights the warm glow of the cake's frosting and highlights the gentle ripples on the water. The overall mood is peaceful and inviting, evoking a sense of serenity and sweetness. The emphasis is on the beauty of the boat and the cake, set against a tranquil backdrop that enhances their simplicity and elegance. The composition is carefully balanced to draw attention to the main subject while highlighting the harmonious setting around it. The result is a scene that exudes calmness, elegance, and pure delight, perfect for evoking a sense of peace and celebration. | 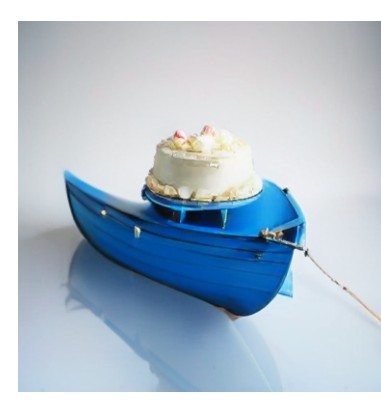 |
| A photo of a train left of a broccoli. | A beautifully vibrant scene features a sleek, polished train gently curving to the side, set against a clear blue sky that highlights its shiny surface. Nearby, a delicate and inviting cluster of broccoli forms a natural, comforting backdrop with its soft green hues contrasting gently with the scene. The overall mood is serene and fresh, evoking a sense of peaceful travel and abundance. Soft natural lighting accentuates the crisp details of the train's polished exterior and the fresh, crisp quality of the broccoli. Together, they create a harmonious balance of elegance and tranquility, capturing the beauty of a bright, natural moment. The scene's composition exudes simplicity and elegance, emphasizing the connection between the train and nature. This inviting and cheerful image invites viewers to embrace the peaceful beauty of a crisp day, perfect for highlighting travel and fresh flavors. | 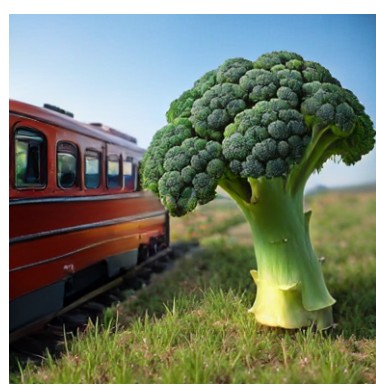 |
| A photo of a sheep right of a book. | A beautiful scene featuring a soft and lively white sheep standing gently next to a sleek, elegant book placed on a simple, clean surface. The lighting highlights the warmth and fresh nature of the setting, emphasizing the natural appearance of the sheep and the book's vibrant cover. The overall tone is serene and welcoming, evoking a sense of calm and peacefulness. The composition emphasizes the contrast between the simplicity of the setting and the intricate details of the book cover and the cozy appearance of the sheep, creating an inviting and harmonious mood. | 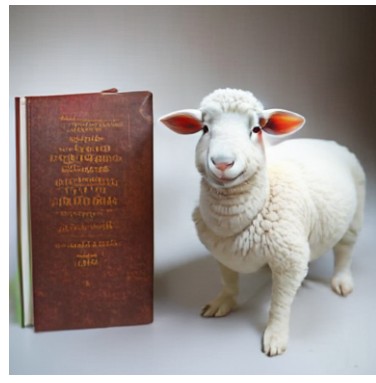 |

*continued from previous page*

| Prompt | CoT | Image |
|--------|-----|-------|
| A photo of a motorcycle left of a sheep. | A sleek motorcycle with a machine-finished body and a stylish design takes a refreshing stand against the natural, sunlit background. Soft natural light enhances the motorcycle's polished surface and highlights its sleek lines. Nearby, a gentle, peaceful sheep exudes an air of calm and serenity. Together, they create a harmonious scene that blends strength and tranquility, evoking a sense of balance and harmony. The colors are warm and inviting, with neutral tones highlighting the harmony between the machines and the gentle animal, set against a natural, softly blurred background. The mood is peaceful and thoughtful, evoking feelings of serenity and balance in an outdoor landscape setting. |  |
| A photo of a donut right of an apple. | A beautifully lit photo showcases a vibrant spot where a plump, ripe apple rests beside a glossy, freshly glazed donut. The colors contrast and complement each other beautifully: the golden hue of the apple, the soft pink reflections on the donut and the shiny glaze icing. Together, they create a delicious and inviting scene, highlighting the contrast between juicy freshness and a sweet, fluffy texture. Soft, natural lighting enhances the smooth surfaces and emphasizes the inviting appearance of the scene, evoking a sense of freshness and sweetness. The background is simple and clean, drawing attention to the main subject without distraction, creating an overall feeling of calm and satisfaction. |  |

Table 12: Visualization Results of REASONGEN-R1 on Complex Attributes Prompts

| Prompt | CoT | Image |
|--------|-----|-------|
| A photo of a brown chair and a purple vase. | A beautifully crafted scene features a cozy brown chair positioned invitingly, exuding warmth and comfort. Soft natural or strategic lighting highlights the rich, warm tones of the chair and creates gentle shadows that enhance its inviting appearance. Beside the chair, a sleek, elegant purple vase adds a touch of sophistication and contrast, complementing the overall mood with its elegant, smooth curves. Together, they create a harmonious and aesthetically pleasing setting, evoking feelings of relaxation and elegance. The overall mood of the image is calm and inviting, with a sense of balance between the warm tones of the chair and the cool elegance of the vase. The scene is carefully composed to highlight the harmony between the elements, emphasizing a sense of balance and tranquility. |  |

*continued from previous page*

| Prompt | CoT | Image |
|---|---|---|
| A photo of a black horse and a blue vase. | A sleek and graceful black horse stands confidently with a sense of power and majesty, set against a calm and soft background that highlights its elegant form. To the side, a serene and elegant blue vase complements the scene, suggesting harmony and balance. Together, the horse and vase create a striking contrast between the dark power of the horse and the delicate beauty of the vase. Soft lighting enhances the smooth texture of the horse's coat and the glossy finish of the vase, emphasizing their rich colors and highlighting their harmonious presence. This quietly powerful image evokes feelings of calmness, elegance, and admiration for the beauty of nature and art combined. | 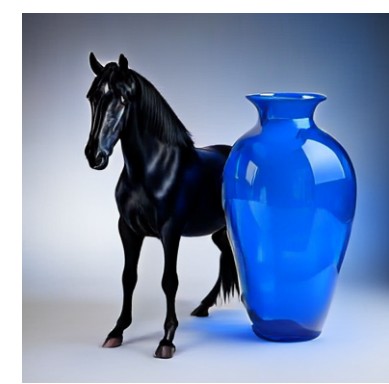 |
| A photo of a green teddy bear and a blue bench. | A charming scene features a vibrant green teddy bear sitting quietly on a bright, expansive surface, its soft texture and gentle expression exuding a sense of comfort and peace. Behind the teddy, a sleek and modern blue structure adds a touch of elegance and sophistication to the composition. Soft, natural lighting highlights the teddy's cheerful presence and emphasizes the crisp blue of the background, creating a balanced and serene atmosphere. Together, the teddy bear and the blue structure evoke a sense of coziness and tranquility, inviting viewers to appreciate the simplicity and harmony of this picturesque moment. | 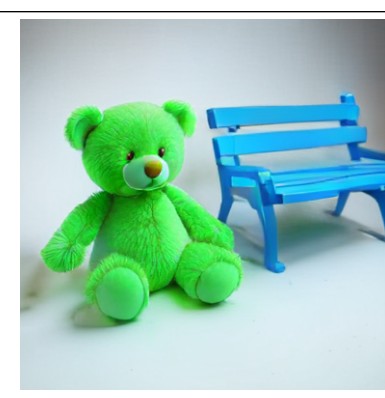 |
| A photo of a purple stop sign and a white potted plant. | A sleek and elegant photo features the natural beauty of a pair showcasing a chic and modern scene. The background is simple and uncluttered, highlighting the gentle contrast between the sleek purple surface of the focal object and the pure white background. Soft, natural lighting accentuates the smooth texture of the surfaces and brings out the crispness of each element, while emphasizing the harmonious balance between the objects. The overall mood is calm and neutral, evoking a sense of sophistication and elegance. Together, the pair captures a perfect harmony that highlights simplicity and understated chic. This carefully composed scene exudes elegance and highlights the elegant contrast between the objects, making it a timeless shot that evokes calm serenity and a sense of balance. | 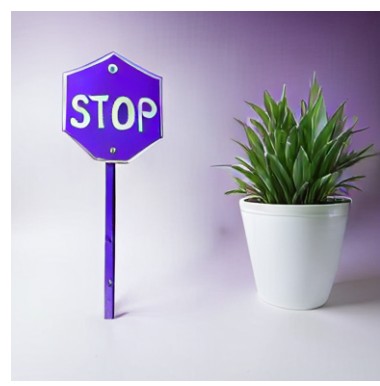 |

*continued from previous page*

| Prompt | CoT | Image |
|---|---|---|
| A photo of a black toaster and a pink bus. | A sleek and elegant black pair of standing appliances creates a sophisticated contrast against a serene, subtle background. A soft, gentle light highlights the smooth curves and shiny surfaces of the modern black objects. Nearby, a charming pink passenger bus stands, blending with the pastel tones of the scene and drawing attention with its vintage design. The composition exudes a blend of elegance and nostalgia, evoking a calm, peaceful mood, with both the black objects and the pink bus harmonizing on the image's surface. Together, they create a balanced and inviting scene that invites viewers to appreciate the subtle contrast and the warmth of the pink and black tones. | 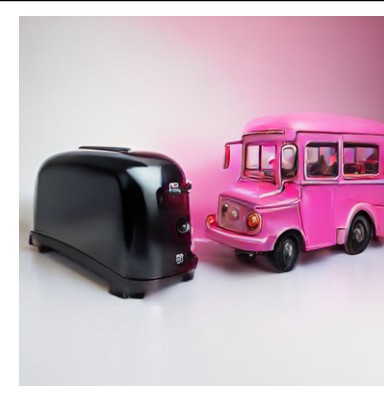 |
| A photo of a green potted plant and a red zebra. | A vibrant scene features a lively green plant with lush, fresh leaves standing proudly beside a striking red horse with a sleek, flowing mane. Together, they create a harmonious balance of natural green and fiery red tones, evoking a sense of contrast and harmony. Soft, natural light highlights the sleek curves of the horse's body and the subtle texture of the leaves, highlighting their smooth surfaces and emphasizing their beauty and elegance. The overall mood is serene and inviting, capturing the spirit of nature and balance in a harmonious outdoor setting that invites viewers to appreciate the beauty and simplicity of this perfect pairing. | 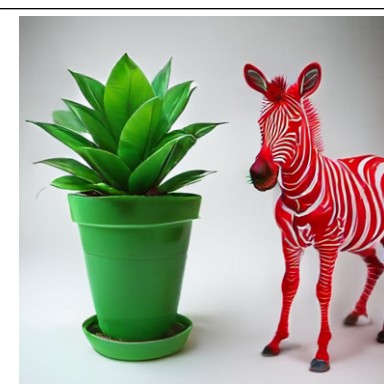 |
| A photo of a red snowboard and a blue bicycle. | A vibrant scene features a sleek, red snowboard standing confidently beside a serene landscape, accentuating a sense of adventure and wonder. Soft, natural lighting highlights the snowboard's bold red hue and the gentle contours of the snowy background. Nearby, a sleek blue bicycle rests gracefully, showcasing its smooth lines and sporty design. Together, these elements evoke a sense of exploration, freedom, and outdoor adventure, inviting viewers to imagine a clear, crisp day in a pristine winter wonderland full of possibilities. The overall mood is peaceful and uplifting, highlighting the beauty of nature and the thrill of outdoor exploration. | 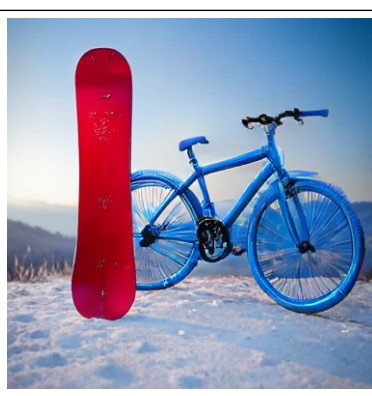 |
| A photo of a white handbag and an orange boat. | A serene scene features a crisp, modern white handbag elegantly displayed beside a bright orange boat, set against a calm, inviting water backdrop. Soft, natural lighting highlights the stark contrast between the pure white of the bag and the warm hue of the boat, evoking a sense of peace and simplicity. The mood is tranquil and inviting, as the subtle details emphasize the simplicity and beauty of the setting. With the reflection of the boat gently shimmering on the water beneath, the photo crafts a harmonious and harmonious atmosphere that combines elegance with a touch of freshness and warmth. | 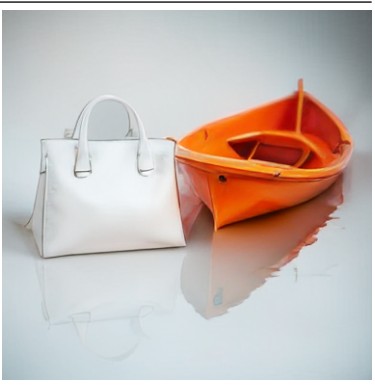 |

*continued on next page*

*continued from previous page*

| Prompt | CoT | Image |
|---|---|---|
| A photo of two sandwichs and two buses. | A delicious sandwich rests invitingly on a clean plate, with the crusty exterior hinting at a flavorful savory filling inside. Nearby, the warm, glowing lights from the bus create a cozy, inviting atmosphere, emphasizing the bustling energy of a city street at night. The scene combines the crispness of streetlights and the gentle hum of the bus engine, shifting the mood to one of peaceful city life after dusk. The harmonious lighting highlights the contrast between the warm glow from the scene and the cool, urban backdrop, evoking a sense of comfort and contentment. No extraneous elements or clutter are present, ensuring the focus remains on the sandwich and the busy street scene, creating a balanced and inviting mood perfect for a quiet evening. |  |

## C  DECLARATION OF LLM USAGE

We used large language models (LLMs) solely for light editing of prose—including wording refinement, grammar correction, and minor clarity improvements—in limited portions of this paper. All LLM-edited text was subsequently reviewed and revised by the authors.

