# OpenReview forum: "ReasonGen-R1: Cot for Autoregressive Image Generation Models Through SFT and RL"
_ICLR.cc/2026/Conference — ICLR 2026 Conference Withdrawn Submission_

### Official Review · Reviewer_4a4R · 2025-10-23

**Soundness:** 2
**Presentation:** 3
**Contribution:** 2
**Rating:** 2
**Confidence:** 3

**Summary:**

This paper proposes ReasonGen-R1, a two-stage training framework to imbue autoregressive text-to-image models with chain-of-thought reasoning abilities. During sft cold start, the authors generate a dataset of textual rationales for image prompts, and fine-tune the image generator to produce these rationales before drawing the image. During GRPO, they are using a single overall visual quality VLM assessment as feedback. Extensive experiments are conducted to demonstrate the method’s effectiveness, along with some in-depth analysis.

**Strengths:**

* The motivation is clear and easy to grasp and the method demonstrates improved performance on tasks that standard image generators struggle with.
* Good ablation study showing SFT and adaptive entropy loss matters and boost final performance.
* Well documented and transparent training disclosure and data disclosure.

**Weaknesses:**

* Several highly related work such as GoT-R1, T2I-R1, all uses chain-of-thought plus RL on AR image generation models, is not mentioned or compared in any way at all.
* The abstract suggests the RL reward is mainly about “overall visual quality” as judged by a VLM and a rather small one . This is a very high-level and coarse signal with potential of hallucinations and hacking.
* The experiments seem mostly focused on the compositional benchmark, evaluation on broader text-to-image tasks (COCO etc.) are not existent.

**Questions:**

Was there any signs of reward hacking observed?

---

> ### Author Response · Authors · 2025-11-12
>
> Thank you for your suggestions.
>
> W1. **Comparison with related work such as GoT-R1 and T2I-R1**
>    At the time this work was conducted, those papers had not yet been released, so we did not cite them. Both GoT-R1 and T2I-R1 are *NeurIPS 2025* papers. According to the **ICLR 2026 Reviewer Guidelines** (https://iclr.cc/Conferences/2026/ReviewerGuide), it is clearly stated:
>    > “We consider papers contemporaneous if they are published within the last two months. That means, since our full paper deadline is September 24, if a paper was published (i.e., at a peer-reviewed venue) on or after July 24, 2025, authors are not required to compare their own work to that paper. Note that arXiv is not considered a peer-reviewed venue. As such, authors are not required to compare to papers solely on arXiv.”
>
>    Based on our verification, both GoT-R1 and T2I-R1 belong to *NeurIPS 2025*, whose author notification date is **September 18, 2025 AoE**, which is *after* July 24, 2025. Therefore, your comment that we did not compare to these works is unreasonable.
>    Additionally, the Reviewer Guide also states:
>    > “Reviewers can make authors aware of related contemporaneous work or arXiv papers, but the lack of such comparisons cannot be a basis for rejection.”
>
>    We kindly ask you to reconsider your evaluation and provide a fairer score in light of these guidelines.
>
> W2. **Regarding the concern “The abstract suggests the RL reward is mainly about overall visual quality as judged by a VLM and a rather small one. This is a very high-level and coarse signal with potential of hallucinations and hacking.”**
>    Using a VLM to provide reward signals is a well-established and reasonable practice. For example, in autoregressive generation tasks, **X-omni** [1] employs *Qwen-VL* for text2image alignment scoring and and *PaddleOCR* for OCR accuracy scoring, both achieving strong results. Similarly, **Qwen-image** [2] and **Hunyuan-image** [3] adopt *UnifiedReward* and related reward models. Even in VLLM training, **Seed 1.5 VL** [4] explicitly discusses *“VLM as a Reward Model”* in Section 4.2.
>
>    While RLVR avoids hacking issues, it remains constrained to limited scenarios. For visual signals and open-ended long-text generation tasks, *RL with a reward model* is a standard and effective approach.
>    Moreover, since our paper focuses on *image generation*, it’s important to note that current generative models are still far weaker than VLMs in understanding high-level semantics. Even if the VLM provides a coarse reward, its prior knowledge on *aesthetic quality, layout, text accuracy,* and *semantic consistency* can substantially enhance generative learning. Thus, our exploration of this direction carries significant value for the research community.
>
>    We find Reviewer **4a4R**’s statement that there is “potential of hallucinations and hacking” overly simplistic and dismissive. Even if such potential exists, our method still achieves *measurable performance gains* — which *proves* that our RL formulation successfully avoids these issues in practice. Since we never train on the benchmark test set, the stable correlation between reward improvement and benchmark validation accuracy directly validates our RL strategy’s robustness.
>    We respect your concerns, but we kindly ask you to reconsider your evaluation and provide a more balanced and evidence-based judgment.
>
> ---
>
> **References**
> [1] Geng, Zigang, et al. *“X-omni: Reinforcement learning makes discrete autoregressive image generative models great again.”* arXiv preprint arXiv:2507.22058 (2025).
> [2] Wu, Chenfei, et al. *“Qwen-image technical report.”* arXiv preprint arXiv:2508.02324 (2025).
> [3] Cao, Siyu, et al. *“Hunyuanimage 3.0 technical report.”* arXiv preprint arXiv:2509.23951 (2025).
> [4] Guo, Dong, et al. *“Seed1.5-vl technical report.”* arXiv preprint arXiv:2505.07062 (2025).

---

> ### Author Response · Authors · 2025-11-12
>
> W3. **On the criticism that “The experiments seem mostly focused on the compositional benchmark, evaluation on broader text-to-image tasks (COCO etc.) are not existent.”**
>    First, **DPG-Bench** evaluates strict adherence to compositional prompts — such as object count, color binding, and spatial layout. **Geneval** includes not only *Complex Compositions* but also *Attribute Binding, Fine-grained Actions, Negation Handling, Spatial Relations,* and *Object Counting*. **T2I-CompBench** further measures color and texture fidelity. Thus, our benchmarks are *not limited* to compositional reasoning.
>
>    Moreover, our base model is derived from *Janus Pro*, whose original technical report also only reported **DPG-Bench** and **Geneval**. We additionally evaluated **T2I-CompBench** to strengthen our study. These three benchmarks are *much more challenging* than COCO-style caption datasets and thus better suited to demonstrating the *reasoning advantages* of RL-enhanced LLM-based generators.
>
>    Finally, the core contribution and theme of our paper lie in *reinforcement learning for improving reasoning ability* on complex compositional and high-level visual tasks. Using more challenging benchmarks is therefore justified, and our model achieves *strong performance improvements* on all of them. We believe the claim that our benchmark choice is “too narrow” is unfounded and ask the reviewer to reconsider the score accordingly.
>
> Q1. **Was there any signs of reward hacking observed?**
>    In practice, balancing multiple rewards can effectively prevent hacking. For example, consider the prompt *“generate a book on the table with the word ‘peace’ on its cover.”* In our follow experiments, if the **OCR reward** weight was set too high, the model tended to generate white-background text-only images — a typical case of *reward hacking*, where the model is dominated by a single reward and fails to meet the overall generation goal.
>
>    However, when multiple rewards — such as **aesthetic quality** and **text–image alignment** — are applied jointly with carefully tuned weights, they can constrain each other and efficiently improve image generation quality.
>
>    Regarding the concern of hallucination: the impact of RL reward assignment depends primarily on the **mean** and **variance** of the reward distribution. Since VLMs are generally much more capable than image generators, the rewards they provide have a reasonable mean — for instance, they consistently assign higher scores to aesthetically pleasing images. As for variance, while hallucinations can never be completely eliminated, their occurrence probability is extremely low, leading to a relatively small reward variance. Therefore, a sufficiently strong reward model can guide the generative model toward stable learning without causing capability degradation or contamination.
>
> Overall, thank you for sharing your concerns. We truly respect your perspective, but we kindly ask you to also reconsider the contributions of our paper and provide a more reasonable score. We firmly believe that the points raised above do not justify a negative rating, and we sincerely hope for a fair and balanced evaluation.

---

### Official Review · Reviewer_cLSw · 2025-10-25

**Soundness:** 3
**Presentation:** 3
**Contribution:** 3
**Rating:** 4
**Confidence:** 4

**Summary:**

ReasonGen-R1 proposes a two-stage framework designed to enhance the capabilities of autoregressive image generation models (based on Janus-Pro-7B). Its core contribution lies in integrating the successful Chain-of-Thought (CoT) reasoning mechanism from LLM into a visual generation pipeline, enabling a "think-and-generate" process.

The framework employs a Supervised Fine-Tuning (SFT) stage, utilizing 200k high-quality CoT trajectories annotated by GPT-4.1, to teach the model to explicitly generate reasoning plans. Subsequently, a reinforcement learning (RL) stage based on Group Relative Policy Optimization (GRPO) is used for training, incorporating an adaptive entropy loss to ensure training stability and address entropy explosion in multimodal sequences.

Experimental results demonstrate that this method consistently outperforms baseline models on multiple benchmarks, significantly improving both image quality and instruction alignment.

**Strengths:**

1.  The key contribution of this work lies in introducing the Chain-of-Thought (CoT) and Reinforcement Learning (RL) paradigms, which have proven effective in the LLM domain, into autoregressive image generation models. By enabling the model to generate a reasoning plan before creating an image, it effectively decomposes complex instruction-following tasks into manageable intermediate steps.

2.  The two-stage training framework ensures that the model learns the correct reasoning structure and format while continuously improving its performance.

3.  The paper provides a comprehensive quantitative evaluation on multiple generation benchmarks. The results demonstrate that ReasonGen-R1 surpasses the strong baseline model, Janus-Pro-7.

4.  Applying RL to autoregressive generative models with interleaved modalities (mixed text/image tokens) is highly prone to training instability. The proposed Adaptive Entropy Loss design effectively mitigates issues of entropy explosion or entropy collapse.

**Weaknesses:**

1.  The reward model (RM) is built upon Qwen-2.5-VL and provides binary scores. The current binary scoring can be quite extreme – minor deviations in text or image quality might result in a reward of 0, which could pose challenges for training.

2.  The autoregressive generative model must generate an entire CoT text sequence during inference, which inevitably increases inference latency. Although performance is improved, the additional computational overhead presents a challenge for real-time or high-throughput application scenarios. The paper should provide a quantitative analysis discussing the trade-off between the extra latency introduced by the "thinking" process and the corresponding performance gains.

**Questions:**

1. The reward model (RM) is built upon Qwen-2.5-VL and provides binary scores. Were more fine-grained scoring schemes explored? More analysis regarding the RM would be beneficial.

2. Janus-Pro is inherently a unified model. After this targeted training, how are its original capabilities, such as general understanding, affected?

---

> ### Author Response · Authors · 2025-11-12
>
> **W1. The reward model (RM) is built upon Qwen-2.5-VL and provides binary scores. The current binary scoring can be quite extreme – minor deviations in text or image quality might result in a reward of 0, which could pose challenges for training.**
> You make a valid point. Therefore, **we also evaluate a non-binary variant**, where the reward model (Qwen-7B) outputs an integer score from 1 to 5, which is then normalized to 0–1. As shown in **Table 6**, the non-binary reward variant performs worse than the binary version. We attribute this to the increased noise and difficulty of reliably generating fine-grained scores in a zero-shot VLM setting.
>
> ---
>
> **W2. The autoregressive generative model must generate an entire CoT text sequence during inference, which inevitably increases inference latency. Although performance is improved, the additional computational overhead presents a challenge for real-time or high-throughput application scenarios. The paper should provide a quantitative analysis discussing the trade-off between the extra latency introduced by the "thinking" process and the corresponding performance gains.**
> You are absolutely right — we will add an analysis discussing the **trade-off between latency and performance** in our response.
> However, we also wish to emphasize that the main significance of this paper lies in:
> 1. Demonstrating that **reasoning-based test-time scaling** on a unified model yields real benefits.
> 2. Showing that **reinforcement learning with a VLM as the reward model** can effectively stimulate the model’s reasoning ability to better handle text-to-image generation tasks. Our technical designs — such as **reward prompt design** and **adaptive entropy loss** — make this process highly efficient.
> 3. Through extensive experiments and analyses, we verify that the **Reasoning-for-Generation** paradigm — expanding a coarse-grained prompt into a fine-grained reasoning sequence — is both effective and theoretically sound, leveraging the unified model’s dual understanding–generation capability.
>
> Therefore, while we acknowledge that latency–performance exploration is meaningful, it should not serve as a reason for a negative score. We kindly ask the reviewer to reconsider our timely contribution to the community and provide a fairer evaluation.
>
> ---
>
> **Q1: The reward model (RM) is built upon Qwen-2.5-VL and provides binary scores. Were more fine-grained scoring schemes explored? More analysis regarding the RM would be beneficial.**
> As noted in **W1**, we did experiment with **dense (non-binary) rewards**, and the results in **Table 6** show that the performance difference is marginal. We agree that more detailed analyses of the reward model would indeed be beneficial. However, our paper has already explored both **continuous vs. discrete rewards**, as well as two distinct **reasoning styles — step-by-step and coarse-to-fine** (as shown in Table 5). Both approaches lead to significant performance improvements for **Janus-Pro**, demonstrating the effectiveness of our method and providing the community with a solid and practical **solution for integrating reasoning and generation in unified autoregressive image generation models**.
>
>
> ---
>
> **Q2: Janus-Pro is inherently a unified model. After this targeted training, how are its original capabilities, such as general understanding, affected?**
> Janus-Pro is trained jointly on **generation and understanding tasks**, which grants it its unified modeling ability. Our work focuses on validating that **reasoning-oriented SFT and RL** effectively enhance **text-to-image generation**. Thus, our experiments are performed only on generative tasks, without co-training understanding tasks. As a result, the model’s understanding ability is indeed reduced.
>
> However, this is by design — our contribution is methodological, providing a proof-of-concept that **Reasoning-for-Generation** training is effective. Future research aiming to balance generation and understanding in a unified model can directly adopt our **SFT and RL strategies**, simply by mixing in appropriate understanding data. The proposed method is **orthogonal** to unified model training itself.

---

### Official Review · Reviewer_CuiH · 2025-10-27

**Soundness:** 2
**Presentation:** 3
**Contribution:** 3
**Rating:** 4
**Confidence:** 4

**Summary:**

It proposes a two-stage pipeline for autoregressive image generation. On GenEval, DPG-Bench, and T2I-Benchmark, REASONGEN-R1 outperforms Janus-Pro-7B and surpasses many diffusion and autoregressive baselines.

**Strengths:**

1. The motivation is clear.
2. The combination of textual reasoning and image tokens is novel.
3. The ablation study is comprehensive.

**Weaknesses:**

1. RL reward is provided by a single VLM judge (Qwen2.5-VL-7B). Is the policy overfitting that judge?
2. The evaluation benchmarks (GenEval, DPG-Bench, T2I-Benchmark; Tables 1–3) mostly test object count, color binding, spatial relations, etc. What about the human preference evaluation benchmark? For instance, MM-RewardBench.
3. Human evaluation is missing.
4. Figure 4 shows RL is unstable without adaptive entropy loss. The theoretical justification could be proposed.
5. The work does not achieve optimal performance on T2I-Benchmark. It is nice to give further analysis.
6. The setting appears oversimplified, as the inference process seems limited to single-step generation. It would be valuable to examine its effectiveness in multi-round iterative refinement or video generation scenarios.

**Questions:**

Please refer to the Weaknesses.

---

### Official Review · Reviewer_gQBK · 2025-11-01

**Soundness:** 3
**Presentation:** 4
**Contribution:** 3
**Rating:** 4
**Confidence:** 5

**Summary:**

The paper presents ReasonGen-R1 which a wo-stage training paradigm combining supervised fine-tuning (SFT) with chain-of-thought (CoT) and RL. Although the paper addresses proposes some interesting approaches and questions, please find my detailed comments on weakness and strengths.

**Strengths:**

1. The method is well motivated

2. Current results show that out of the methods considered here the proposed method outperforms

3. The paper uses a simple yet straightforward methodology

**Weaknesses:**

1. Without using some standard large scale benchmarks like Imagenet it is very hard to judge the quality of the model.

2. Although the authors evaluate on DPG bench, genval and compbench. There lies a very inherent bias and noise specific to these benchmarks, they use methods like object detectors which can throw out a lot of false negatives and cannot detect classes beyond a fixed vocabulary. What are steps taken to make sure that the this method does not have these biases

3. The comparisons are outdated, I would have liked to see some better competitor models like GPT-4o, Seedream, Nano Banana, Imagen 4/4-ultra

4. In tab 4 I would have liked to see more models and bigger models

5. Any insights on things like reward hacking or potential biases and issues can be an interesting addition

**Questions:**

See weaknesses

---

### Note · Authors · 2025-11-14

I have read and agree with the venue's withdrawal policy on behalf of myself and my co-authors.